# Run-up, Inundation, and Sediment Characteristics of December 22 2018 Indonesia Sunda Strait Tsunami

Wahyu Widiyanto[1,2], Shih-Chun Hsiao[1], Wei-Bo Chen[3], Purwanto B. Santoso[2], Rudy T. Imananta[4], Wei-Cheng Lian[1]

[1]Department of Hydraulic and Ocean Engineering, National Cheng Kung University, Tainan, 701, Taiwan
[2]Department of Civil Engineering, Universitas Jenderal Soedirman, Purwokerto, 53122, Indonesia
[3]National Science and Technology Center for Disaster Reduction, New Taipei City, 23143, Taiwan
[4]Meteorological, Climatological and Geophysical Agency (BMKG), Jakarta, 10720, Indonesia

*Correspondence to*: Shih-Chun Hsiao (schsiao@mail.ncku.edu.tw)

**Abstract.** A tsunami caused by a flank collapse of the southwest part of the Anak Krakatau volcano occurred on December 22, 2018. The tsunami affected the coastal areas located at the edge of Sunda Strait, Indonesia. To gain an understanding of the tsunami event, field surveys were conducted a month after the incident. The surveys included measurements of run-up height, inundation distance, tsunami direction, and sediment characteristics at 20 selected sites. The survey results revealed that the run-up height reached 9.2 m in Tanjungjaya and inundation distance 286.8 m was found at Cagar Alam, part of Ujung Kulon National Park. The tsunamis propagated radially from Anak Krakatau and reached the coastal zone with direction between 25° and 350° from North. Sediment samples were collected at 27 points in tsunami deposits with a sediment thickness of 1.5-12.7 cm. The average distance of area with significant sediment deposits and deposit limit from the coast are 45% and 73% of the inundation distance respectively. Sand sheets were sporadic, highly variable, and highly influenced by topography. Grain sizes in the deposit area were finer than those at their sources. The sizes ranged from fine sand to boulders, with medium sand and coarse sand being dominant. All sediment samples had a well sorted distribution. An assessment of the boulder movements indicates that the tsunami run-up had minimum velocities of 4.0-4.5 ms$^{-1}$.

## 1 Introduction

A tsunami took place in Sunda Strait on December 22, 2018, at 22:00 Western Indonesia Time (+7 UTC). It shocked the local residents because it came without any warning signs, such as earthquake shocks. The source of the tsunami was the Anak Krakatau volcano, a sea mountain in the middle of Sunda Strait. The southwestern slope of the mountain experienced a landslide that resulted in the movement of sea water, which propagated to land in the form of a tsunami wave. When the tsunami reached land, its large energy caused a lot of damage and casualties. Records obtained from the Indonesian National Disaster Management Agency (Indonesian: Badan Nasional Penanggulangan Bencana, BNPB) show 430 deaths, 1015 collapsed houses, and a lot of other damage (e.g., seawalls, revetments, jetties, boats, and cars). The affected areas in Banten Province include Pandeglang and Serang Districts and those in Lampung Province include the regencies of South Lampung, Tanggamus, and Pesawaran.

Sunda Strait is home to the Krakatau (or Krakatoa) volcano. It is famous for the 1883 Krakatau eruption, which caused a 30-m tsunami that led to 36,000 fatalities and affected the Earth's climate and weather for several weeks, as reported by Verbeek (1884). The 1883 eruption of Krakatau and the resulting tsunami have been widely discussed (e.g., Yokoyama 1987; Camus *et al.* 1992; Maeno and Imamura 2011; Paris *et al.* 2014). A young volcano called Anak Krakatau (Child of Krakatau) appeared above sea level in 1929. It grew to 338 m above sea level in September 2018. This very active volcano was the source of the tsunami discussed in the present study.

The generation of the tsunami that occurred on December 22, 2018, in Sunda Strait was triggered by the collapse of a flank in the southwest part of the Anak Krakatau volcano. Satellite imagery shows that the area of the body of the marine volcano that was lost was 64 hectares; the collapsed volume was estimated to be 150-180 million m³ (Kasbani, 2018). As a result of the collapse of some of the volcano's body, the volcano's height decreased from 338 to 110 m above sea level. The tsunami caused

by the collapse of the Anak Krakatau flank was investigated by Giachetti et al. (2012). They used a numerical model to simulate an unstable flank collapse in the southwest part of Anak Krakatau. A hypothetical volume of 280 million m³ produces a wave with an initial height of 43 m on Sertung, Panjang, and Rakata islands that then spreads to the beaches in the western part of Java Island, including Merak, Anyer, Carita, Panimbang, and Labuhan, and Sumatra Island, reaching Bandar Lampung City. The actual area affected by the December 22 event is consistent with their model, but different in magnitude due to the use of

the worst case scenario in the simulation. The numerical modelling of the December 2018 Anak Krakatau tsunami was performed by Heidarzadeh et al. (2020) while Muhari et al. (2019) conducted field surveys of this event to record tsunami run-up along the coasts of Sunda Strait.

Tsunamis in Sunda Strait are of great concern because the strait is important both locally and globally. It connects the two main islands of Java and Sumatra, whose population accounts for 79% of Indonesia's population (BPS-Statistics Indonesia,

2019). About 6.9 million people live in the coastal area of the strait in Banten Province and Lampung Province (BPS-Statistics of Banten Province, 2019; BPS-Statistics of Lampung Province, 2019). The strait, between Merak and Bakauheni, is the busiest inter-island crossing in Indonesia, with 17,824,392 passengers and 4,218,548 vehicles in 2018 (Dirjen Perhubungan Darat, 2019). The strait is also an international route for large ships. It is the second-most crowded waterway after Malacca Strait, with 70,000 vessels a year passing it (Soeriaatmadja, 2016). There are three industrial regions at the edge of the strait, namely

Cilegon, Serang, and Tanggamus. There is also a special economic zone in this region, namely Tanjung Lesung. The beaches in the strait are a tourist destination. There are two UNESCO world heritage sites across from each other; one at the western tip of Java Island (Ujung Kulon National Park) and the other at the southern tip of Sumatra Island (Bukit Barisan Selatan National Park). Bandar Lampung, which has a population of 1 million, is the provincial capital and faces the strait directly. Jakarta, the capital of the Republic of Indonesia, is relatively close to the strait.

Post-tsunami field surveys were conducted to obtain data for future mitigation and development activities in the region. The surveys began exactly a month after the tsunami, January 22, and ended on January 28, 2019. Our team included people from Indonesia and Taiwan. We carried out measurements of the run-up height and inundation distance of the tsunami. In addition, we also identified flow directions and sediment deposits caused by the tsunami.

## 2 Study Area

The tsunami has had a serious impact on life in the surrounding area. The affected area covers the coastal area on the western tip of Java Island (Banten Province) and the southern tip of Sumatra Island (Lampung Province). Banten Province covers two districts, namely Serang and Pandeglang. Lampung Province covers South Lampung Regency and the provincial capital of Lampung, namely Bandar Lampung. Post-tsunami field surveys were conducted in these areas. We selected 20 sites for observation and measurement (Table 1 and Fig. 1). These sites are located along 140 km of coast on Java Island and 80 km of

coast on Sumatra Island. Sites 13 and 14 were reached by boat because of the difficult land route.

Land use at these sites includes housing, agriculture, tourism, and a national park. Sites 1, 2, 4, 7, 8, 9, and 10 (Karangsuraga, Pasauran, Pejamben, Tanjungjaya 1, and Tanjunglesung 1-3) have houses mixed with hotels, resorts, and villas. Sites 13, 15, 17, and 20 (Kertajaya Sumur, Bumi Waras, Wayurang 2, and Kunjir) have high-density housing. Sites 3, 6, 7, and 16 (Sukarame, Mekarsari, Tanjungjaya 1, and Wayurang 1) have agricultural land. Site 14 is a protected national park with limited

access. Although this site has no residents, it is very important to review it because it includes the only Javan rhino (*Rhinoceros sondaicus*) habitat in the world. The Javan rhino is one of the most threatened mammals on earth, with a population of less than 100. The distribution of the rhino indicates that tsunamis are a significant risk to the species in the area (Setiawan et al.,

2018). Each site has a different beach profile. Most sites are natural beaches. Sites 2, 4, 8, 9, and 10 (Pasauran, Pejamben, and Tanjunglesung 1-3) have coastal structures (e.g., sea walls) and sites 5, 17, and 20 (Sukamaju, Wayurang 2, and Kunjir) have revetments.

**Figure 1.** Locations of field surveys. Sunda Strait lies between Java Island and Sumatra Island in Indonesia. Red dots and numbers attached show survey site locations and survey site numbers respectively. Inundation distances and run-up heights for each site are shown in the table. Base maps are from BIG (Badan Informasi Geospasial, Indonesia).

**Table 1.** Field survey sites and measurements

## 3 Method

Measurements of run-up and inundation were conducted using conservative terrestrial surveying methods with optical and laser devices (e.g., total stations, handheld GPS devices, and laser distance meters). We measured run-up and inundation based on coastline at the time of survey. Run-up were corrected to calculate heights above sea level because the tide level at the time of actual tsunami was different from tidal level at the time of surveys. We use WXTide software version 4.7 for correcting elevation. Elevation values of each survey site were corrected with the nearest tidal gauge available. We used 3 station in Ciwandan, Labuhan and Teluk Betung, for corrections. The maximum run-up and inundation limits are based on remaining tsunami trail marks at measurement locations. The tracks were in the form of debris, fallen trees, plants that change color, and damage to buildings. The observed damage to buildings and structures was caused by the tsunami because there were no other

causes, such as the earthquake sand liquefaction in the 2018 Sulawesi tsunami (Widiyanto et al., 2019). In addition, information regarding inundation limits and highest run-up was also obtained from eyewitnesses. IOC Manuals and Guides No. 37 (1998 and 2014) and field survey reports (Maramai and Tinti 1997; Farreras 2000; Matsutomi *et al.* 2001; Fritz and Okal 2008) were used as guidelines for the implementation of this field survey.

Sediment samples were collected from selected points at measurement locations that could be in the swash zone, nearshore,

berm, or deposit areas (Table 2 and Fig. 2). The measurement of deposit thickness in sandy sheets was carried out by digging a number of shallow holes. The measured thickness was considered to be near the maximum thickness. This method was qualitative and subjective because tsunami deposits are discontinuous, sporadic, and scattered over a flooded area. Sand sheets deposited on land vary greatly due to the influence of sedimentary sources and topography. A pit was made at each selected point to observe layers suspected to have been produced by the tsunami. We took only one sample at each pit for laboratory

testing and did not take vertical samples at intervals of 1 cm, as done by some researchers (Gelfenbaum and Jaffe 2003; Hawkes *et al.* 2007; Srinivasalu *et al.*2007; Srisutam and Wagner 2010), because a detailed analysis was not our focus, particularly the number of tsunami waves and the vertical variation of sediment. The grain size of the samples obtained from the field was tested using ASTM standard sieve analysis. In addition, we investigated boulder movement at four sites.

**Figure 2.** Aerial photographs from Google Earth of transects including highest run-up point and deposit pit.

## 4 Run-up

The run-up was measured by determining the height difference between the highest point of sea water rise onto land and the coastline. Run-up is influenced by the characteristics of the ground surface and slope. The measurement results from our field

surveys show that run-up ranged from 1 to 9 m (Table 1 and Fig. 1). The values in the table and figure have been corrected for tide to obtain elevation from sea level at time of tsunami. A run-up height of about 1 m was found in many locations, at which

no damage was found. The highest run-up was found at the Tanjung Jaya 2, Cagar Alam, and Kunjir sites, with heights of 9.2, 7.6, and 7.8 m respectively. Site Tanjungjaya 2 is located in Cipenyu Beach. Muhari et al. (2019) reported maximum run-up height of 13 m in area around Tanjungjaya/Cipenyu Beach as well. This value is significant different with our maximum run-up since we measure in flat valley part of Cipenyu Beach while Muhari et al. measured in hilly coast of Cipenyu Beach. The

Cagar Alam (sanctuary) site is part of the Ujung Kulon National Park. This site has a flat topography and a dense forest. The guard post and water police office were completely destroyed by the tsunami. The Kunjir site, which is densely inhabited, had the second most victims after the Sumur site. This site is located on Sumatra Island about 38 km from the tsunami source. The Tanjung Jaya 2 site is a private resort with many tourists. The topography is relatively flat but suddenly rises at a distance of about 250 m from the coastline due to a long hill. A large boulder moved by the tsunami was found at this site.

Our surveyed run-up heights are compared with published tide gauge records of Heidarzadeh et al. (2020). Three tide gauges mentioned in the article (Ciwandan, Marina Jambu and Panjang) are used. Maximum amplitudes at Ciwandan, Marina Jambu, and Panjang are 1.15 m, 2.8 m, and 1.25 m respectively. Ciwandan tide gauge is used to evaluate run-up heights at site Karangsuraga, Pasauran, Sukarame and Pejamben. Marina Jambu tide gauge is used to evaluate run-up heights at sites Sukamaju, Karangsari, Tanjungjaya 1, Tanjunglesung (1,2,3), Tanjungjaya 2, Banyuasih, Kertajaya Sumur, and Cagar Alam.

Besides, Panjang tide gauge is used to evaluate sites of Bumiwaras, Wayurang 1, Wayurang 2, Kotaguring, Sukaraja, and Kunjir. It is indicated that averaged run-up heights of each site associated with the tide gauge are 4 times, 1.15 times, and 3.1 times larger than maximum amplitude at the Ciwandan, Marina Jambu, and Panjang respectively. The sites are relatively far from the tide gauge.

## 5 Inundation

The distance from the run-up point to the coastline is defined as the inundation distance (IOC Manuals and Guides No. 37, 2014). This distance can be easily obtained using a distance measurement instrument or GPS. We used total station for this purpose. The coastlines elevation in our survey were corrected with tide elevation of several tide gauge in Sunda Strait. The results of our field measurements show that the inundation distance ranged from 10 to 290 m (Table 1 and Fig. 1). The wave with an inundation distance of 11 m and a run-up of 1.2 m at site 15 (Bumi Waras) was not felt by the population. This site

was chosen to represent the area of Bandarlampung City, which is the capital of Lampung Province. This city has a population of 1 million (2018) and must thus develop tsunami mitigation strategies. The longest inundation distance was found at site 14 (286.8 m), in the sanctuary, which also had the high run-up. At this site, measurements were made near the mouth of a small river. Long inundation distances may be caused by relatively flat topography with relatively few obstacles. Tsunami may also travel faster through a stream channel. A relatively long inundation (263.1 m) was also found at Tanjungjaya 2, a site with the

highest run-up. This site is in the form of a valley plain with a small stream. Slope in the valley is relatively flat and suddenly changing steeply in hilly areas within 250-300 m from coastline. The run-up point we recorded is located on the slope change from mild to steep. These mild and steep areas have slopes of approximately 0.025 and 0.06, respectively. Local people call this area as Cipenyu Beach. This is a sandy beach flanked by cliffs or hilly beaches. Fortunately, not many people live around this site other than at a resort complex, which suffered severe damage.


## 6 Tsunami Wave Direction

The tsunami spread out from its source on the Anak Krakatau volcano to the beaches at the edge of Sunda Strait. To determine the direction of the tsunami that arrived at the beach, we obtained information from eyewitnesses. The tsunami hit at night and thus its arrival was difficult to see. Fortunately, it hit during a full moon period, so that there was some light. In addition to

eyewitness accounts, we obtained evidence in the field related to the direction of the tsunami propagation. The evidence was

in the form of fallen tree trunks, sloping vegetation and shrubs, damaged buildings, and building components carried away by the flow (Fig. 3).

**Figure 3. Evidence of tsunami direction. Arrows show the direction of tsunami flow on the ground surfaces.**

Our survey results show that the direction of the tsunami propagation was radial from the source (Fig. 4). The tsunami travelled east on the coast between Anyer and Labuhan (sites 1-6). In the vicinity of Tanjunglesung (sites 7-13), the tsunami was directed to the southeast, and in part of Ujung Kulon National Park (Site 14), the tsunami was directed southward. The tsunami was directed to the North and slightly to the northeast on Sumatra Island (sites 15-19). The westward tsunami toward the Tanggamus area was relatively small and insignificant. We did not include this area in the survey. The smaller magnitude of the tsunami to the west is likely due to obstruction by the island of Sertung and bathymetry factors. The Anak Krakatau mountain avalanche had a southwest direction, but the tsunami in this direction had no impact on human life because it leads to the open sea (the Indian Ocean), with increasing depth from the tsunami source. Tsunami wave direction from North arrived in coastal area is given in Table 1 for the field survey sites. The direction ranges from 25° to 350° from North that it indicates radially propagation of the tsunami wave.

**Figure 4.** Direction of tsunami propagation, the tsunami spread radially from its source. Numbers in red dots and images are survey site numbers. Base maps are from BIG (Badan Informasi Geospasial, Indonesia) and images are from Google Earth ©.

## 7 Sediment Characteristics

### 7.1 Tsunami Deposits

Prehistoric (paleo-) tsunamis have been identified from sediment deposits in several studies (Atwater 1992; Dawson and Shi 2000; Peters, Jaffe and Gelfenbaum 2007). Sediment deposits can be used to explain and reconstruct significant tsunami events (Dawson *et al.* 1995; Van Den Bergh *et al.* 2003). The present study attempts to describe the impact of a recent tsunami on sediment movement around the coastal area. The tsunami carried sediment from the coast inland. However, not all sites that we measured had significant sediment deposits. Only places with a sufficient source of material had clearly observable sediment deposits (Fig. 5). The survey sites used for sediment samples are shown in Fig. 2. Sediment deposits generally do not spread evenly and continuously but are separated at certain locations, which allow it to settle. Topography controls sediment deposits, for instance, there are more sediment deposits at ground surface depressions.

The best location for the observation of tsunami sediment is about 50-200 m inland from the coastline (Srisutam and Wagner, 2010) or about 50-400 m inland (Moore et al., 2006), as used for the 2004 Sumatra-Andaman tsunami. In this study, the 12 deposit pits were 9-195 m from the shoreline. Four deposit pits were less than 50 m from the shoreline (Fig. 6). Three of them were at sites 1, 2, and 7, respectively, due to the short inundation and beach scarp. Another was at site 13 (Kertajaya Sumur), where high-density housing blocked the sediment transport and created a deposit at short distance from the shoreline.

**Figure 5.** Tsunami sediment deposit, test pits were made to measure the deposit thickness.

The interpretation of tsunami magnitude, especially run-up and inundation based on tsunami deposits, is challenging (Dawson and Shi 2000). However, the relationship between deposits and run-up or inundation is still not convincing because of the high variability of tsunami deposits in terms of thickness and location. Soulsby et al. (2007) proposed a mathematical model for reconstructing tsunami run-up from sedimentary characteristics. The run-up distance for sediment is related to the run-up distance for water as:

$$R_s = \frac{R_w}{1+\alpha\gamma} \tag{1}$$

where $R_s$ is the maximum distance inland for sediment deposition, $R_w$ is the run-up limit for water, $\alpha$ is as shown in Eq. 2, and $\gamma$ is a comparison factor between uprush time and total uprush plus backwash time.

$$\alpha = \frac{w_s T}{H} \tag{2}$$

where $w_s$ is the settling velocity, $T$ is a period from the time of first wetting to final drying of inundated ground, and $H$ is the tsunami height. Fig. 6 shows the distance of measured sediment deposition and water run-up compared to the distance of theoretical sediment deposition calculated using Eq. 1 and Eq. 2, the results are in good agreement. However, Sukarame, Tanjungjaya 1 and Cagar Alam do not fit well. The three location have morphological conditions may not ideal for applying the theoritical approach. Sukarame has beach scarp and tsunami flows across a stream around 90 m from coastline. Tanjungjaya 1 has also beach scarp and there is a sea wall, although not so high, that may block the sediment movement. Eventhough Tanjungjaya 1 has abundant material, low amplitude tsunami caused a few sand transport. Cagar Alam has a relative larger stream than Sukarame. In addition, Cagar Alam has dense vegetation since it is a national park.

As it can be seen in Fig. 6, the distance of area with significant sediment deposits caused by the tsunami from the coast (deposit pit distance) varied in the range of 9-195 m (average: 82 m) from the shoreline or 10%-70% (average: 45%) of the inundation distance. Moreover, the limits where tsunami sediments were found that can still be seen with the naked eye (deposit limit distance) are around 11-244 m (average 122 m) from the coastline or 30%-90% (73% on average) of the inundation distance. Fig. 6 also presents the elevation of the deposit pit and deposit limit to the runup elevation. The average elevation of deposit pit and deposit limit are 43% and 72% of the runup elevation respectively.

**Figure 6.** Positions of deposit pits and deposit limit compared to inundation distance and run-up elevation. Theoritical approach for tsunami deposit only available for distance not for elevation.

The sediment samples were tested for gradations in the laboratory. The results for each site are shown in Fig. 8. The sediment in the deposit areas on land is generally finer than at the sources at the beach (nearshore or swash zone). The characteristics of the sediment are discussed in Sect. 7.3. The sediment deposition thickness varied greatly from one point to another at the survey sites. We chose a test pit with significant thickness for sampling. The thickness was likely near the maximum sediment thickness in the area.

## 7.2 Boulder Movement

Coastal boulder accumulation is usually associated with high-energy events (tsunamis, hurricanes, or powerful storms). A characteristic of many tsunamis is their ability to deposit boulders across the coastal zone (Dawson and Shi, 2000). Extreme storms also have the ability to deposit boulders (Morton, Gelfenbaum and Jaffe 2007; Richmond *et al.* 2010). The interpretation of boulders is difficult along coasts where both storms and tsunamis have occurred. We identified boulders moved by a tsunami wave and run-up at three survey sites based on information from eyewitnesses. Eyewitnesses said that these boulders were in new positions after the tsunami. In addition, from the physical criteria given by Morton et al. (2007) and Paris et al. (2010), it was most likely that the boulders were moved by the tsunami. One of criteria by Morton et al. (2007) we found in this site is a relatively thin (average < 25 cm) bed composed of normally graded sand consisting of a single structureless bed or a bed with only a few thin layers. Sediment thickness around the boulder is very thin. Paris et al. (2010) reported regarding boulder and fine sediment transport and deposition by the 2004 tsunami that most of the sediments deposited on land came from offshore, from fine sands to coral boulders, and with very high values of shear velocity (>30 cm/s). The boulder we found came from nearshore and a part of the boulder was submerged. We estimate that high shear velocity should occure to transport it. It was most possible by 22 December 2018 tsunami. At about the time of the tsunami event, a tropical cyclone called Kenanga formed

in the Indian Ocean about 1400 km from Sunda Strait. Kenanga had a speed of 75 km/h and was active from December 15 to 18, 2018 (Prabowo, 2018). The influence of this cyclone was weak in the coastal zone, and thus it was unlikely to have moved the boulders.

The largest boulder, measuring 2.7 m in diameter (10.4 tons), was found at site 11 (Tanjungjaya 2), as shown in Fig. 7a. This boulder moved from its original point in the swash zone to 82 m inland. Other smaller stones were scattered around it. In Tanjunglesung (sites 8, 9, and 10), pebbles, cobbles, and small boulders (25-50 cm) were scattered up to 40 m from the coastline. At site 4, the seawall built to protect a hotel and villas was partly destroyed and moved ashore. A seawall chunk, measuring 1 m ×1 m × 4.2 m (9.5 tons), made from rubble mound and mortar moved as far as 30 m from its place of origin (Fig. 7b). Other smaller chunks were also moved.

**Figure 7.** (a) Boulder that moved around 82 m inland in Tanjungjaya 2 and (b) an element of seawall that moved around 30 m inland in Pejamben.

The characteristics of the boulders moved by a tsunami can be used to estimate the associated flow velocities. For instance, the 2004 tsunami had flow velocities of 3-13 m/s. This tsunami drove a 7.7-ton calcareous boulder 200 m and an 11-ton coral boulder as far as 900 m (Paris et al., 2010). The velocity was calculated as:

$$u = \sqrt{\left(\frac{2\mu m g}{C_d A_n \rho_w}\right)} \tag{3}$$

where $\mu$ is the friction coefficient, $m$ is the boulder mass (kg), $g$ is the gravitational acceleration, $C_d$ is the drag coefficient, $A_n$ is the area of the boulder projected normal to the flow (m$^2$), and $\rho_w$ is the density of sea water (kgm$^{-3}$). The velocities were calculated from Equation 3 to be $u \geq 4.5$ ms$^{-1}$ and $u \geq 4.0$ ms$^{-1}$ for the 10.4-ton (Fig. 7a) and 9.4-ton (Fig. 7b) boulders, respectively.

## 7.3 Sand Size Statistics

The results of sieve analysis, namely sand grain size distributions, are shown as a cumulative distribution curve of sand grain size. Fig. 8 shows the cumulative distribution curve for 12 sites. From the curve, various diameter values were determined, including $d_{95}$, $d_{84}$, $d_{50}$, $d_{16}$, and $d_5$. From these diameters, other statistics can be determined, namely the mean, standard deviation, skewness, and kurtosis (Table 2). The mean can be used for grain size classification. The standard deviation is a measure of range that shows the uniformity of a sand sample. A perfectly sorted sample will have sand of the same diameter, whereas poorly sorted sand will have a wide size range. Beach sand size distributions with a standard deviation of $\leq 0.5$ are considered well sorted, and those with a standard deviation of $\geq 1$ are assumed to be poorly sorted. Skewness occurs when the sand size distribution is not symmetrical. A negative skewness value indicates that the distribution is tending to the value of small phi (large grain size). Kurtosis determines the peakedness of the size distribution. The normal distribution has a kurtosis value of 0.65. If the distribution is more diffuse and wider than the normal distribution, the kurtosis value will be less than 0.65 (Dean and Dalrymple, 2004).

From our results, the mean values show that medium and coarse were the dominant types of sand in the sample. Very coarse sand, granular sand, and pebbles were found at the Tanjungjaya 2 and Sukarame sites. Fine and very fine sand was also identified at several sites. The range of grain sizes found in the study area depends on the available source material. Wentworth classification was used to assess the grain size. All samples had negative standard deviations, indicating that they had well sorted distributions.

**Figure 8.** Sediment grain size results from sieve analysis for various sites.

7 of the 10 samples taken from the swash zone had negative skewness, which indicated a large phi value and an erosive tendency in the zone. The numbers of samples with positive and negative skewness were similar (7 and 6, respectively). Some deposit samples were taken at a distance of less than 50 m from the coastline, which may still be an erosive environment.

The kurtosis of the tsunami sediment indicates that grain size distributions were flat to peaked distribution. Generally, the major sources of tsunami sediment are swash zones and berm/dune zone sands, where coarse to medium sands are dominant. A minor source of tsunami sediment is the shoreface, where fine to very fine sands are dominant. However, for a coastal area where the shoreface slope is mild, the major source of tsunami sediment is the shoreface. Table 2 provides kurtosis values from which distribution of sediment range from very platykurtic to very leptokurtic.

## 8 Conclusion

We selected 20 sites on Java Island and Sumatra Island to observe the impact of the December 2018 tsunami, which was caused by a mass movement of an Anak Krakatau volcano flank. The survey results revealed that the run-up height ranged from 1 to 9 m, the inundation distance was 10 to 290 m, and the direction of the tsunami was between 25° and 350°. The highest run-up (9.2 m) was found at site Tanjungjaya 2. The longest inundation distance (286.8 m) was found and site Cagar Alam which contains a forest area, part of a national park, and a UNESCO heritage site. Sediment samples were taken from 27 points in tsunami deposits with a sediment thickness of 1.5-12 cm. The distance of area with significant sediment deposits caused by the tsunami from the coast varied in the range of 9-195 m (average: 82 m) from the shoreline or 10%-70% (average 45%) of the inundation distance. Meanwhile, deposit limit distances are around 11-244 m (average 122 m) from the coastline or 30%-90% (average 73%) of the inundation distance. The average elevation of deposit pit and deposit limit are 43% and 72% of the runup elevation respectively. Sediment material larger than coarse sand (granular sand, pebbles, cobbles, and boulders) was found at several locations. The largest boulder has a diameter of 2.7 m and a weight of 10.4 tons. From the boulder movement, the tsunami velocity at the ground surface is estimated to be more than 4.5 ms$^{-1}$. Sand size statistics are also given in this report. The sediment grain size ranged from very fine sand to boulders, with medium sand (diameter: 0.25-0.5 mm) and coarse sand (diameter: 0.5-1.0 mm) being dominant. All sediment samples tested in the laboratory has a well sorted distribution, indicating that the grain sizes were relatively uniform.

## Data availability

Data can be made available by the authors upon request.

## Author contributions

WW and SCH designed the field survey. WW and WCL conducted the field survey in the disaster area. WW and SCH wrote original draft. SCH managed funding acquisition. WBC corresponded with Taiwanese and Indonesian government offices. PBS and RTI provided early information and mobilized surveyors and measurement tools. All authors contributed to the discussion and interpretation of the results.

## Competing interests

The authors declare that they have no conflict of interest.

## Acknowledgment

The authors would like to thank the Geomatics Laboratory and the Soil Mechanics Laboratory at the Civil Engineering Department, Universitas Jenderal Soedirman, Purwokerto, Indonesia. We would also like to thank Ujung Kulon National Park and Tanjung Lesung Special Economic Zone for conducting surveys in their areas. We are grateful to Mr. Sumantri from

Tanjunglesung Special Economic Zone and Mr. Budi Prasetyo from Carita Resort for accompanying the tsunami impact review and providing detailed chronological information on the tsunami, especially in Tanjunglesung and Carita Beach, respectively. We appreciate Ms. Sanidhya Nika Purnomo for discussion on geographical information.

**Financial Support**

This study was funded by the Ministry of Science and Technology, Taiwan, under grant MOST 108-2221-E-006-087-MY3 and MOST 109-2217-E-006-002-MY3.

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

**Table 1.** Field survey sites and measurements

| Site | Site name | Measurement time (dd/mm/yy) (+7 UTC) | Coordinates | | Uncorrected inundation distance (m) | Corrected inundation distance (m) | Uncorrected run-up height (m) | Corrected run-up height (m) | Tsunami direction from North (°) | Sediment sample numbers |
|---|---|---|---|---|---|---|---|---|---|---|
| | | | Long. (°) | Lat. (°) | | | | | | |
| 1 | Karangsuraga | 25/01/19 16:30:38 | -6.148434 | 105.854718 | 84.0 | 84.3 | 4.7 | 4.7 | 82 | S-01, S-02 |
| 2 | Pasauran | 25/01/19 14:59:16 | -6.202289 | 105.836179 | 111.0 | 103.2 | 5.8 | 5.4 | 84 | S-03, S-04 |
| 3 | Sukarame | 24/01/19 16:44:32 | -6.261508 | 105.830448 | 204.7 | 179.4 | 4.1 | 3.6 | 93 | S-05, S-06, S-07 |
| 4 | Pejamben | 24/01/19 14:43:46 | -6.316783 | 105.831298 | 200.8 | 181.5 | 5.0 | 4.6 | 105 | - |
| 5 | Sukamaju | 24/01/19 18:11:48 | -6.390917 | 105.825965 | 125.8 | 89.2 | 2.2 | 1.5 | 110 | - |
| 6 | Mekarsari | 25/01/19 18:36:37 | -6.520795 | 105.758381 | 123.8 | 86.4 | 2.3 | 1.6 | 115 | S-08 |
| 7 | Tanjungjaya 1 | 27/01/19 08:34:50 | -6.509201 | 105.673902 | 60.6 | 35.7 | 1.8 | 1.1 | 210 | S-09, S-10 |
| 8 | Tanjunglesung 1 | 23/01/19 12:30:08 | -6.480980 | 105.659513 | 149.5 | 131.0 | 3.4 | 3.0 | 135 | S-11, S-12, S-13 |
| 9 | Tanjunglesung 2 | 23/01/19 11:33:12 | -6.480914 | 105.654575 | 202.2 | 194.9 | 3.6 | 3.5 | 128 | - |
| 10 | Tanjunglesung 3 | 23/01/19 10:42:46 | -6.481270 | 105.652097 | 177.1 | 176.7 | 2.4 | 2.4 | 133 | S-14, S-15, S-16 |
| 11 | Tanjungjaya 2 | 23/01/19 14:54:20 | -6.504584 | 105.642052 | 284.2 | 263.1 | 10.1 | 9.2 | 120 | S-17, S-18, S-19 |
| 12 | Banyuasih | 23/01/19 15:55:39 | -6.600539 | 105.621017 | 170.3 | 148.1 | 2.1 | 1.2 | 142 | - |
| 13 | Kertajaya Sumur | 26/01/19 18:47:58 | -6.656894 | 105.583253 | 134.7 | 115.9 | 5.0 | 4.2 | 150 | S-20, S-21 |
| 14 | Cagar Alam | 26/01/19 13:36:21 | -6.676569 | 105.378788 | 292.2 | 286.8 | 7.8 | 7.6 | 180 | S-22, S-23, S-24 |
| 15 | Bumi Waras | 28/01/19 10:21:37 | -5.459514 | 105.262263 | 10.0 | 11.0 | 1.1 | 1.2 | 350 | - |
| 16 | Wayurang 1 | 28/01/19 13:26:34 | -5.723070 | 105.582329 | 181.6 | 196.9 | 4.2 | 4.5 | 30 | S-25, S-26, S-27 |
| 17 | Wayurang 2 | 28/01/19 14:28:27 | -5.745806 | 105.587961 | 81.3 | 86.3 | 4.2 | 4.4 | 34 | - |
| 18 | Kotaguring | 28/01/19 14:55:30 | -5.800583 | 105.584414 | 29.1 | 31.4 | 2.6 | 2.8 | 25 | - |
| 19 | Sukaraja | 28/01/19 15:22:13 | -5.833699 | 105.626956 | 64.4 | 68.3 | 2.4 | 2.6 | 40 | - |
| 20 | Kunjir | 28/01/19 15:34:50 | -5.834768 | 105.642150 | 207.1 | 210.4 | 7.7 | 7.8 | 47 | - |

**Table 1.** Field survey sites and measurements

| Sample | Date (dd/mm/yy) | Time (+7 UTC) | Lat. (°S) | Long. (°E) | Village | Subdistrict | Regency | Zone | Deposit thickness (cm) | Median $D_{50}$ (mm) | $\Phi_{50}$ | Mean | Std. Dev. | Skew-ness | Kurto-sis | Remarks |
|---|---|---|---|---|---|---|---|---|---|---|---|---|---|---|---|---|
| S-01 | 25/01/19 | 16:30:38 | -6.148187 | 105.854549 | Karangsuraga | Cinangka | Serang | swash zone | - | 0.650 | 0.621 | 0.883 | -1.368 | -0.191 | -0.864 | coarse sand, well sorted, very platykurtic |
| S-02 | 25/01/19 | 16:45:44 | -6.148195 | 105.854774 | Karangsuraga | Cinangka | Serang | deposit | 12 | 0.330 | 1.599 | 1.698 | -0.698 | -0.141 | 0.504 | medium sand, well sorted, very platykurtic |
| S-03 | 25/01/19 | 15:02:22 | -6.202145 | 105.835194 | Pasauran | Cinangka | Serang | swash zone | - | 0.200 | 2.322 | 0.007 | -2.177 | 1.063 | 0.536 | fine sand, well sorted, very platykurtic |
| S-04 | 25/01/19 | 15:19:01 | -6.201856 | 105.835437 | Pasauran | Cinangka | Serang | deposit | 3 | 0.500 | 1.000 | 0.652 | -1.237 | 0.281 | 0.603 | coarse sand, well sorted, very platykurtic |
| S-05 | 24/01/19 | 16:44:32 | -6.260923 | 105.828634 | Sukarame | Labuhan | Pandeglang | nearshore | - | 3.000 | -1.585 | -1.243 | -1.757 | -0.195 | 0.463 | granular, well sorted, very platykurtic |
| S-06 | 24/01/19 | 16:49:13 | -6.261013 | 105.828977 | Sukarame | Labuhan | Pandeglang | swash zone | - | 0.500 | 1.000 | 0.921 | -0.769 | 0.103 | 0.599 | coarse sand, well sorted, very platykurtic |
| S-07 | 24/01/19 | 16:54:01 | -6.261021 | 105.829628 | Sukarame | Labuhan | Pandeglang | deposit | 6.3 | 0.500 | 1.000 | 0.268 | -1.368 | 0.461 | 0.579 | coarse sand, well sorted, very platykurtic |
| S-08 | 25/01/19 | 18:54:59 | -6.520795 | 105.758381 | Mekarsari | Panimbang | Pandeglang | deposit | 12.7 | 0.330 | 1.599 | 1.656 | -0.740 | -0.076 | 0.497 | medium sand, well sorted, very platykurtic |
| S-09 | 27/01/19 | 08:34:50 | -6.509219 | 105.674029 | Tanjungjaya 1 | Panimbang | Pandeglang | swash zone | - | 0.240 | 2.059 | 1.994 | -0.743 | 0.087 | 1.516 | fine sand, well sorted, very leptokurtic |
| S-10 | 27/01/19 | 08:35:35 | -6.509201 | 105.672903 | Tanjungjaya 1 | Panimbang | Pandeglang | deposit | 2 | 0.650 | 0.621 | 0.128 | -1.561 | 0.316 | 0.550 | coarse sand, well sorted, very platykurtic |
| S-11 | 23/01/19 | 12:22:32 | -6.479796 | 105.658553 | Tanjunglesung 1 | Panimbang | Pandeglang | swash zone | - | 0.220 | 2.184 | 2.266 | -0.792 | -0.103 | 0.790 | fine sand, well sorted, platykurtic |
| S-12 | 23/01/19 | 12:15:53 | -6.480013 | 105.658580 | Tanjunglesung 1 | Panimbang | Pandeglang | deposit | 3 | 0.170 | 2.556 | 2.791 | -1.005 | -0.233 | 0.715 | fine sand, well sorted, platykurtic |
| S-13 | 23/01/19 | 12:32:23 | -6.480799 | 105.659432 | Tanjunglesung 1 | Panimbang | Pandeglang | deposit | 1.5 | 0.260 | 1.943 | 1.808 | -0.749 | 0.181 | 1.730 | medium sand, well sorted, very leptokurtic |
| S-14 | 23/01/19 | 10:52:19 | -6.480284 | 105.652178 | Tanjunglesung 3 | Panimbang | Pandeglang | swash zone | - | 0.400 | 1.322 | 1.630 | -0.844 | -0.365 | 0.722 | medium sand, well sorted, very leptokurtic |
| S-15 | 23/01/19 | 11:04:00 | -6.480438 | 105.652278 | Tanjunglesung 3 | Panimbang | Pandeglang | deposit | 6.8 | 0.410 | 1.286 | 1.117 | -0.883 | 0.192 | 1.936 | medium sand, well sorted, very leptokurtic |
| S-16 | 23/01/19 | 11:53:24 | -6.481134 | 105.652333 | Tanjunglesung | Panimbang | Pandeglang | deposit | 3.2 | 0.380 | 1.396 | 1.619 | -0.703 | -0.317 | 2.008 | medium sand, well sorted, very leptokurtic |
| S-17 | 23/01/19 | 14:28:44 | -6.503221 | 105.639871 | Tanjungjaya 2 | Panimbang | Pandeglang | nearshore | - | 4.200 | -2.070 | -2.276 | -1.074 | 0.191 | 0.365 | pebble, well sorted, very platykurtic |
| S-18 | 23/01/19 | 14:38:08 | -6.502949 | 105.640477 | Tanjungjaya 2 | Panimbang | Pandeglang | swash zone | - | 1.200 | -0.263 | -0.026 | -1.460 | -0.163 | 0.332 | very coarse sand,well sorted,very platykurtic |
| S-19 | 23/01/19 | 14:51:05 | -6.504377 | 105.641338 | Tanjungjaya 2 | Panimbang | Pandeglang | deposit | - | 0.380 | 1.396 | 1.429 | -0.893 | -0.037 | 0.588 | medium sand, well sorted, very platykurtic |
| S-20 | 26/01/19 | 18:40:59 | -6.655898 | 105.583705 | Kertajaya | Sumur | Pandeglang | swash zone | - | 0.950 | 0.074 | 0.465 | -1.272 | -0.037 | 0.778 | coarse sand, well sorted, platykurtic |
| S-21 | 26/01/19 | 14:42:08 | -6.656034 | 105.583687 | Kertajaya | Sumur | Pandeglang | deposit | 2 | 0.220 | 2.184 | 1.826 | -0.911 | 0.393 | 1.131 | medium sand, well sorted, leptokurtic |
| S-22 | 26/01/19 | 12:27:00 | -6.673185 | 105.379727 | Cagar Alam | Sumur | Pandeglang | nearshore | - | 0.220 | 2.184 | 1.800 | -0.937 | 0.410 | 1.654 | medium sand, well sorted, very leptokurtic |
| S-23 | 26/01/19 | 13:14:17 | -6.674877 | 105.379122 | Cagar Alam | Sumur | Pandeglang | swash zone | - | 0.380 | 1.396 | 1.494 | -0.828 | -0.119 | 1.071 | medium sand, well sorted, mesokurtic |
| S-24 | 26/01/19 | 14:03:24 | -6.675483 | 105.378968 | Cagar alam | Sumur | Pandeglang | deposit | 7.5 | 0.120 | 3.059 | 3.020 | -0.624 | 0.063 | 0.861 | very fine sand, well sorted, platykurtic |
| S-25 | 28/01/19 | 14:04:46 | -5.724157 | 105.581636 | Wayurang 1 | Kalianda | S. Lampung | nearshore | - | 0.190 | 2.396 | 2.370 | -0.814 | 0.031 | 0.697 | fine sand, well sorted, platykurtic |
| S-26 | 28/01/19 | 13:48:51 | -5.723089 | 105.581996 | Wayurang 1 | Kalianda | S. Lampung | swash zone | - | 0.190 | 2.396 | 2.398 | -0.660 | -0.003 | 0.995 | fine sand, well sorted, mesokurtic |
| S-27 | 28/01/19 | 14:04:46 | -5.724157 | 105.581636 | Wayurang 1 | Kalianda | S. Lampung | deposit | 12.3 | 0.170 | 2.556 | 2.959 | -0.930 | -1.718 | -2.248 | fine sand, well sorted, very platykurtic |

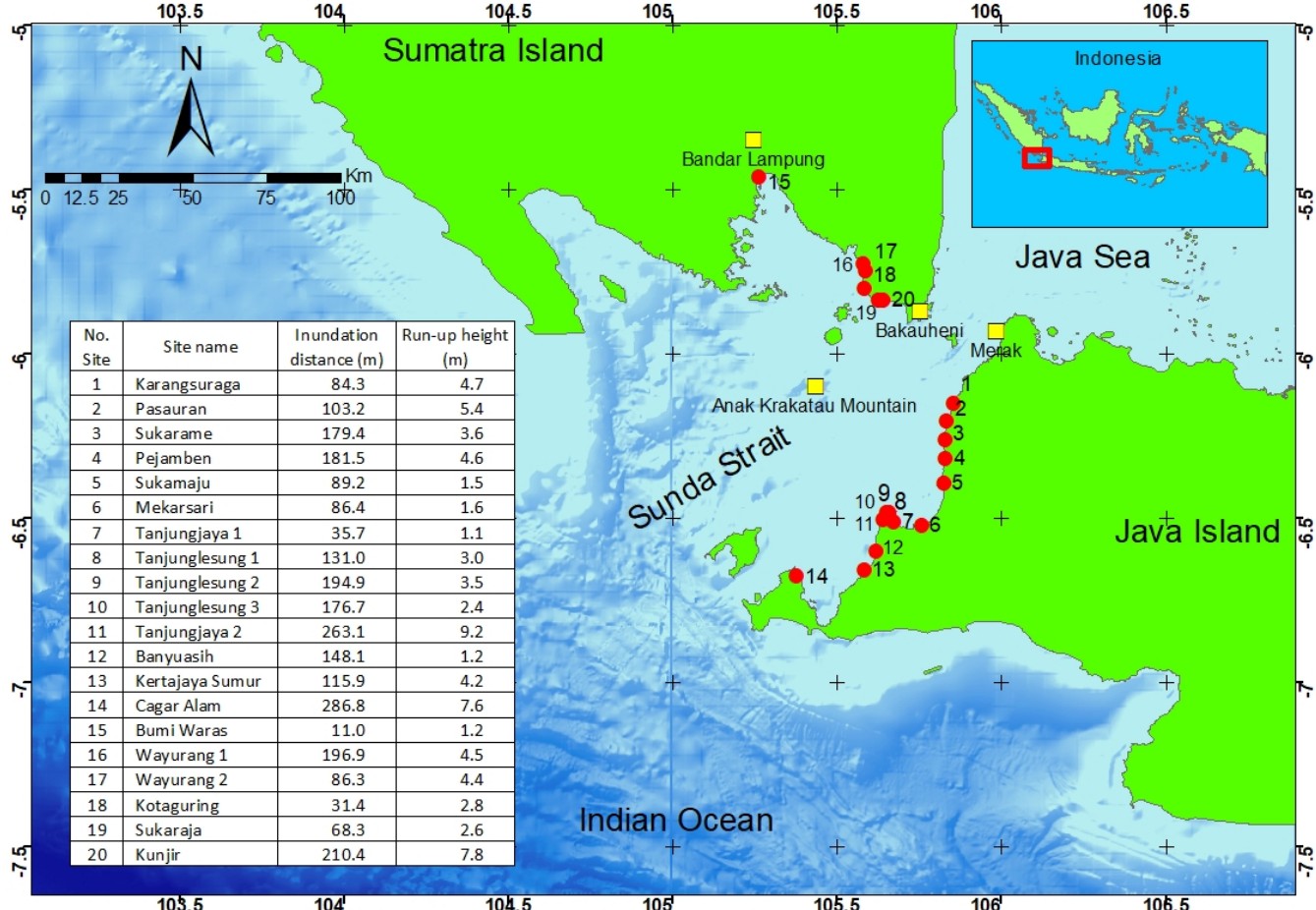

| No. Site | Site name | Inundation distance (m) | Run-up height (m) |
|---|---|---|---|
| 1 | Karangsuraga | 84.3 | 4.7 |
| 2 | Pasauran | 103.2 | 5.4 |
| 3 | Sukarame | 179.4 | 3.6 |
| 4 | Pejamben | 181.5 | 4.6 |
| 5 | Sukamaju | 89.2 | 1.5 |
| 6 | Mekarsari | 86.4 | 1.6 |
| 7 | Tanjungjaya 1 | 35.7 | 1.1 |
| 8 | Tanjunglesung 1 | 131.0 | 3.0 |
| 9 | Tanjunglesung 2 | 194.9 | 3.5 |
| 10 | Tanjunglesung 3 | 176.7 | 2.4 |
| 11 | Tanjungjaya 2 | 263.1 | 9.2 |
| 12 | Banyuasih | 148.1 | 1.2 |
| 13 | Kertajaya Sumur | 115.9 | 4.2 |
| 14 | Cagar Alam | 286.8 | 7.6 |
| 15 | Bumi Waras | 11.0 | 1.2 |
| 16 | Wayurang 1 | 196.9 | 4.5 |
| 17 | Wayurang 2 | 86.3 | 4.4 |
| 18 | Kotaguring | 31.4 | 2.8 |
| 19 | Sukaraja | 68.3 | 2.6 |
| 20 | Kunjir | 210.4 | 7.8 |

**Figure 1.** Locations of field surveys. Sunda Strait lies between Java Island and Sumatra Island in Indonesia. Red dots and numbers attached show survey site locations and survey site numbers respectively. Inundation distances and run-up heights for each site are shown in the table. Base maps are from BIG (Badan Informasi Geospasial, Indonesia).

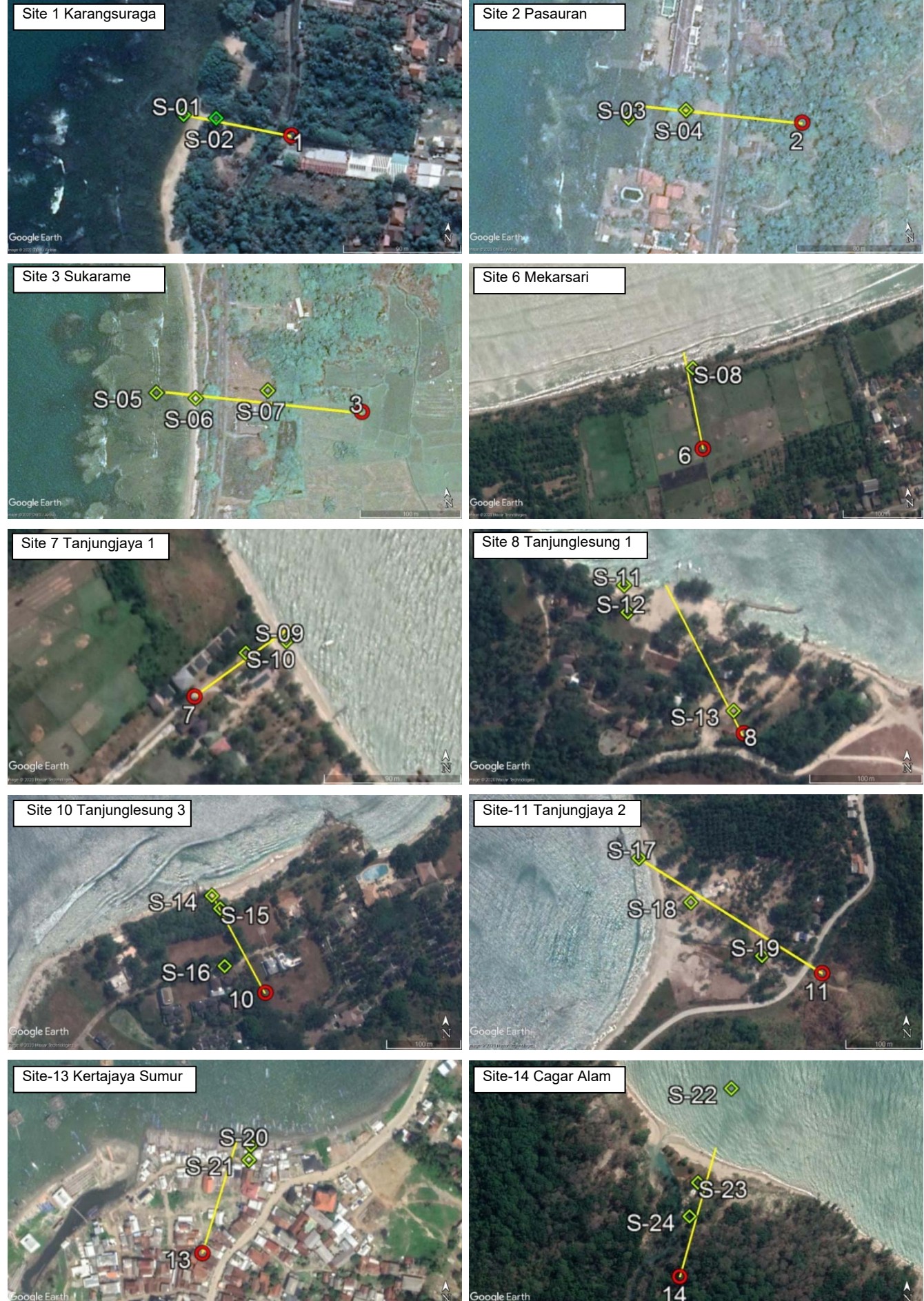

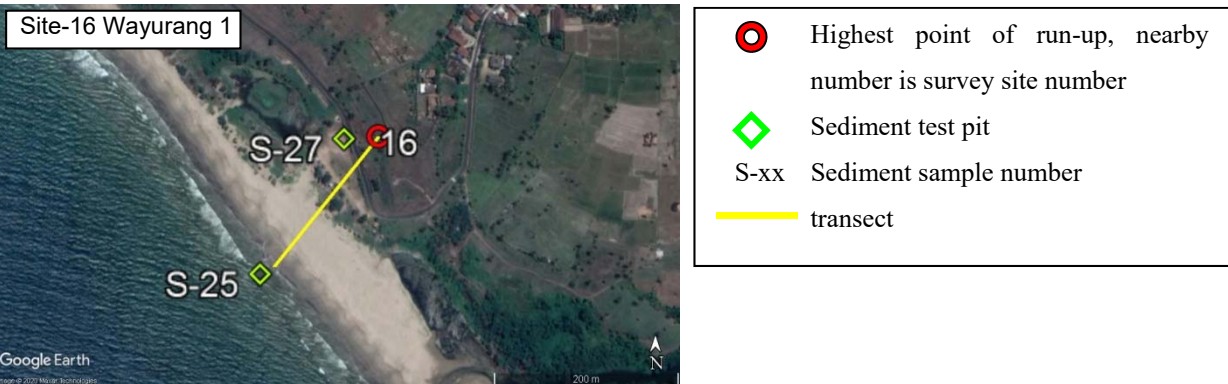

**Figure 2.** Aerial photographs from Google Earth of transects including highest run-up point and deposit pit.

25

30

35

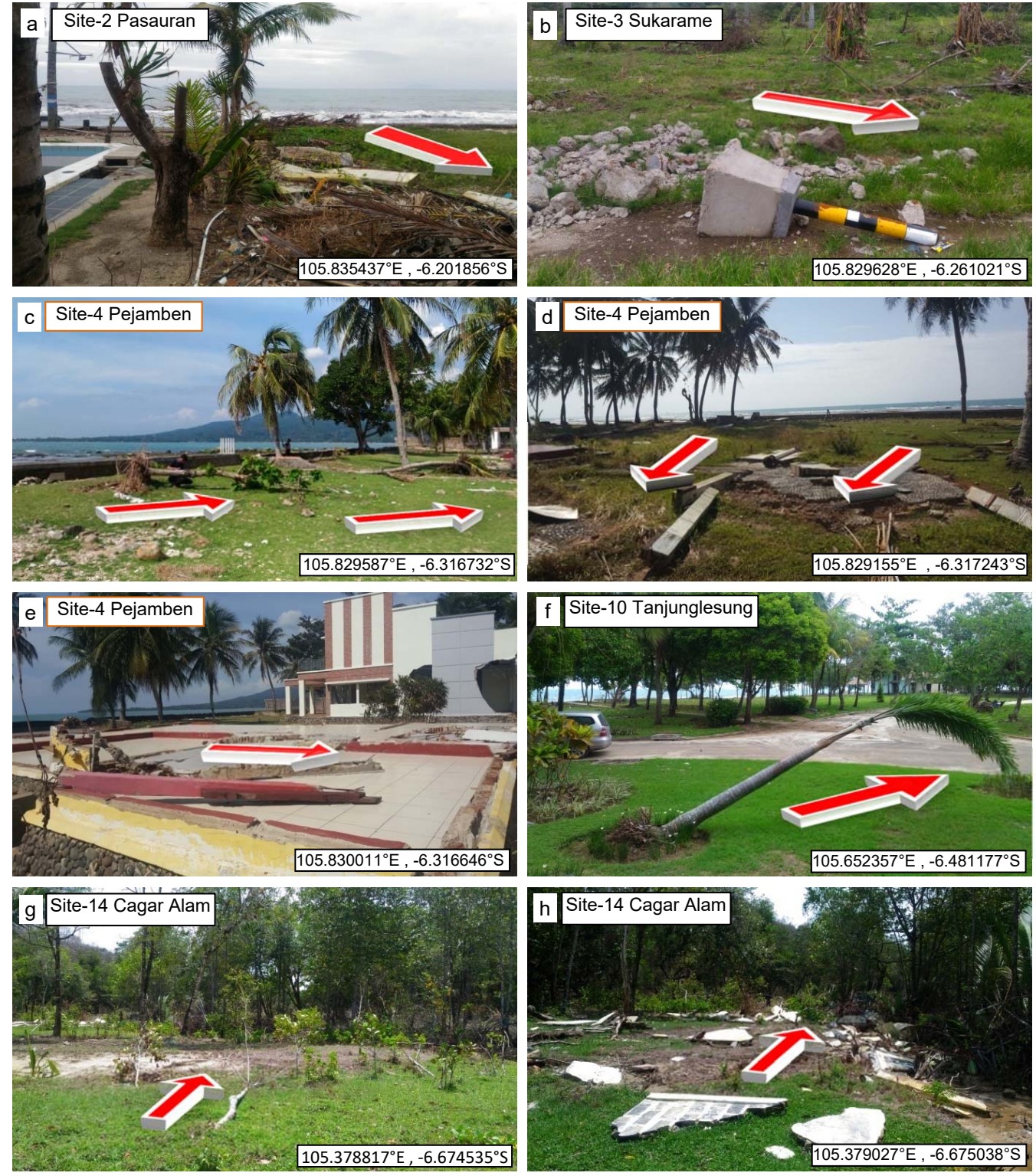

**Figure 3.** Evidence of tsunami direction. Arrows show the direction of tsunami flow on the ground surfaces.

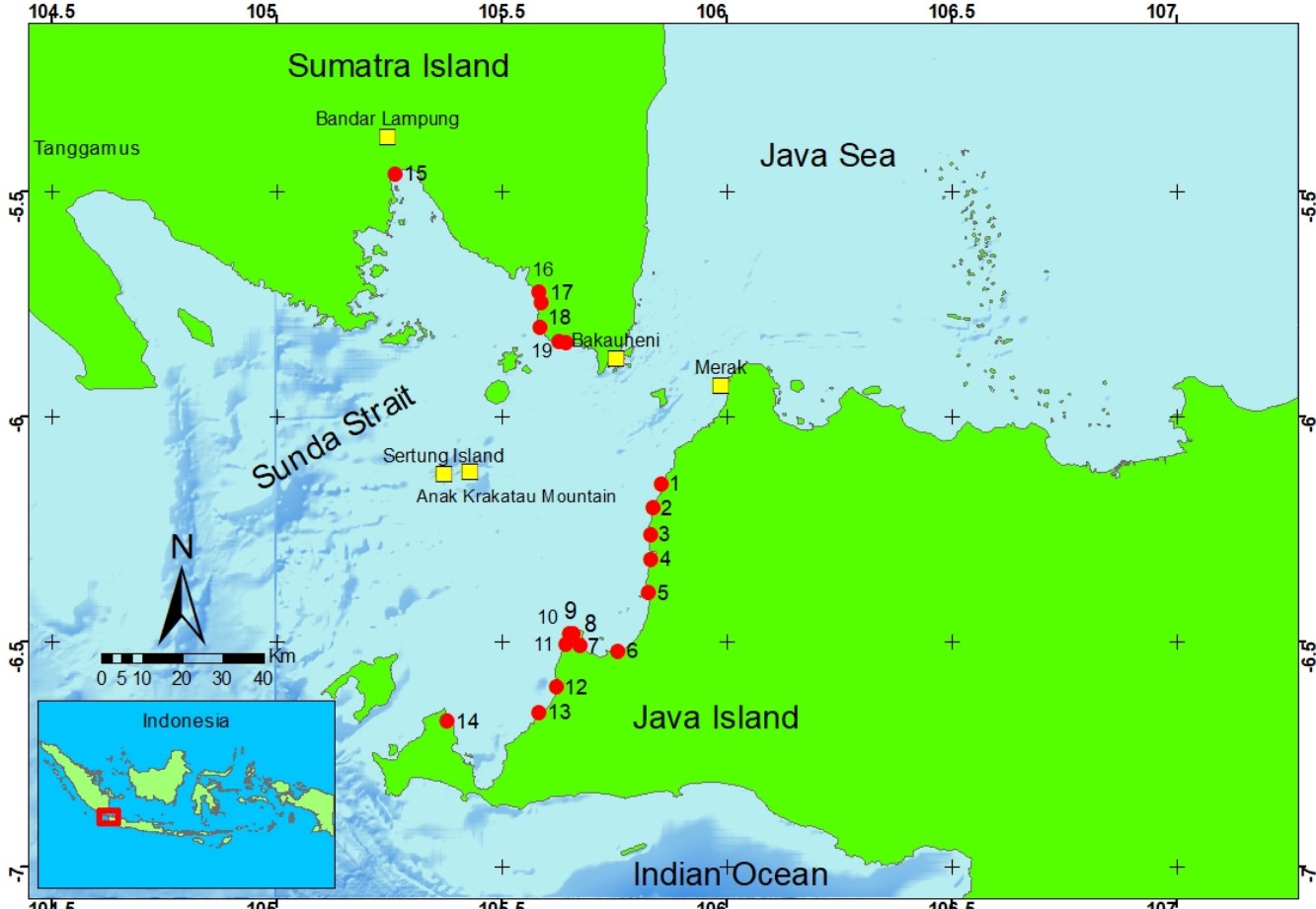

**Figure 4.** Direction of tsunami propagation, the tsunami spread radially from its source. Numbers in red dots and images are survey site numbers. Base maps are from BIG (Badan Informasi Geospasial, Indonesia) and images are from Google Earth ©.

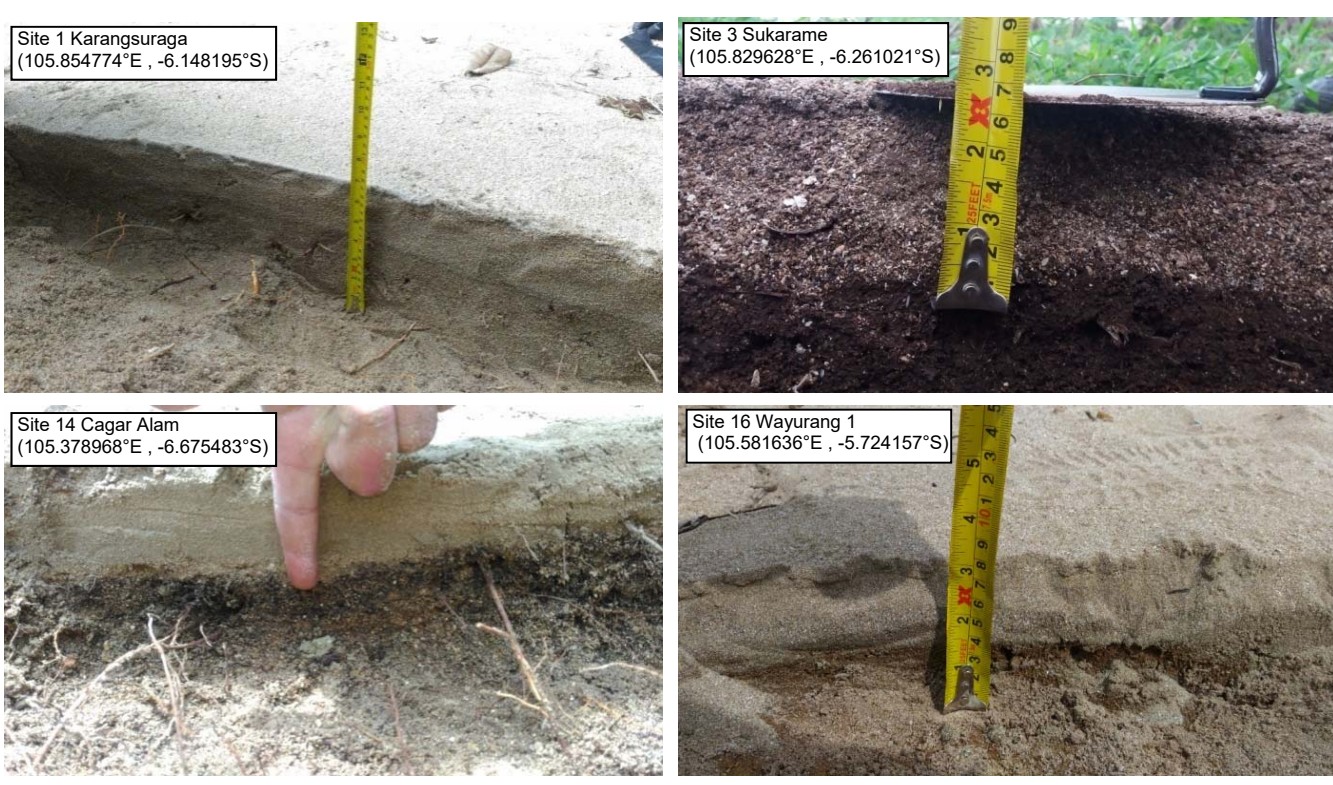

**Figure 5.** Tsunami sediment deposit, test pits were made to measure the deposit thickness.

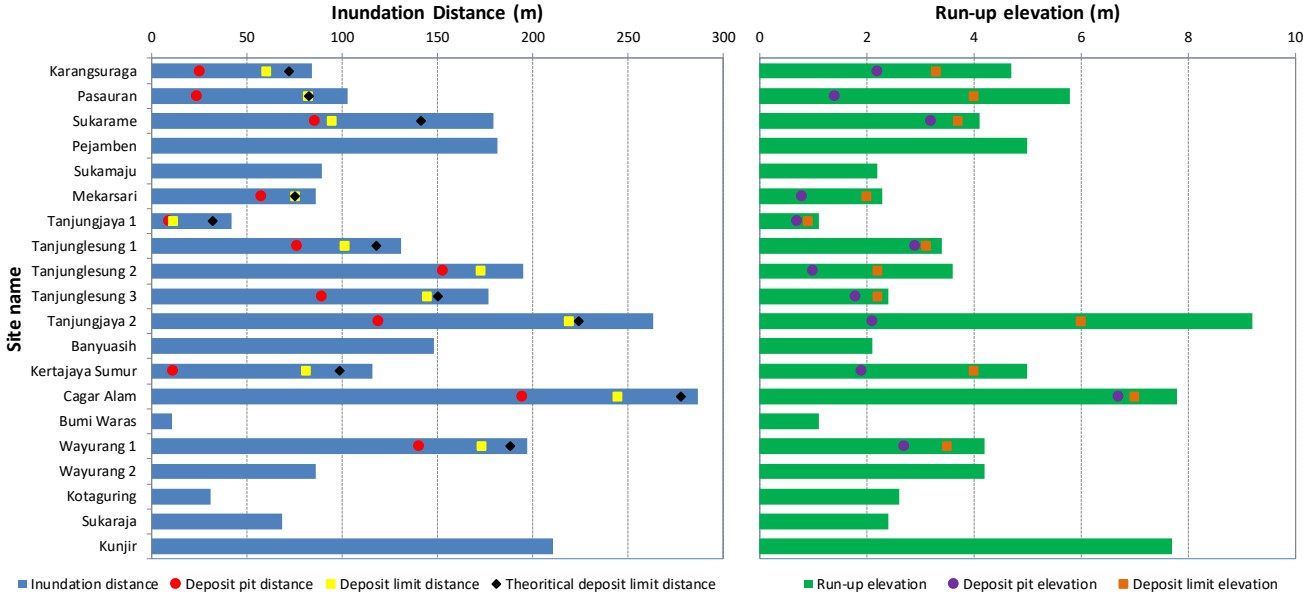

**Figure 6.** Positions of deposit pits and deposit limit compared to inundation distance and run-up elevation. Theoritical approach for tsunami deposit only available for distance not for elevation.

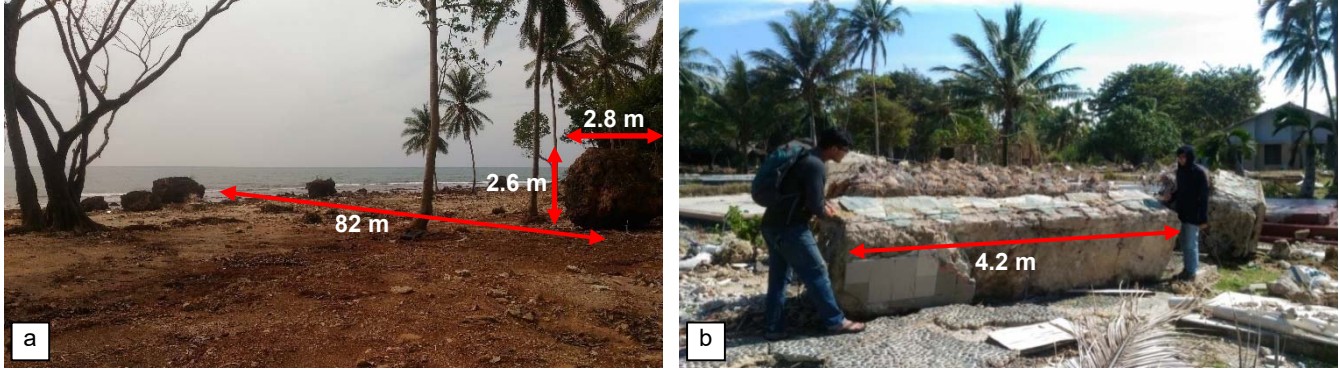

**Figure 7.** (a) Boulder that moved around 82 m inland in Tanjungjaya 2 and (b) an element of seawall that moved around 30 m inland in Pejamben.

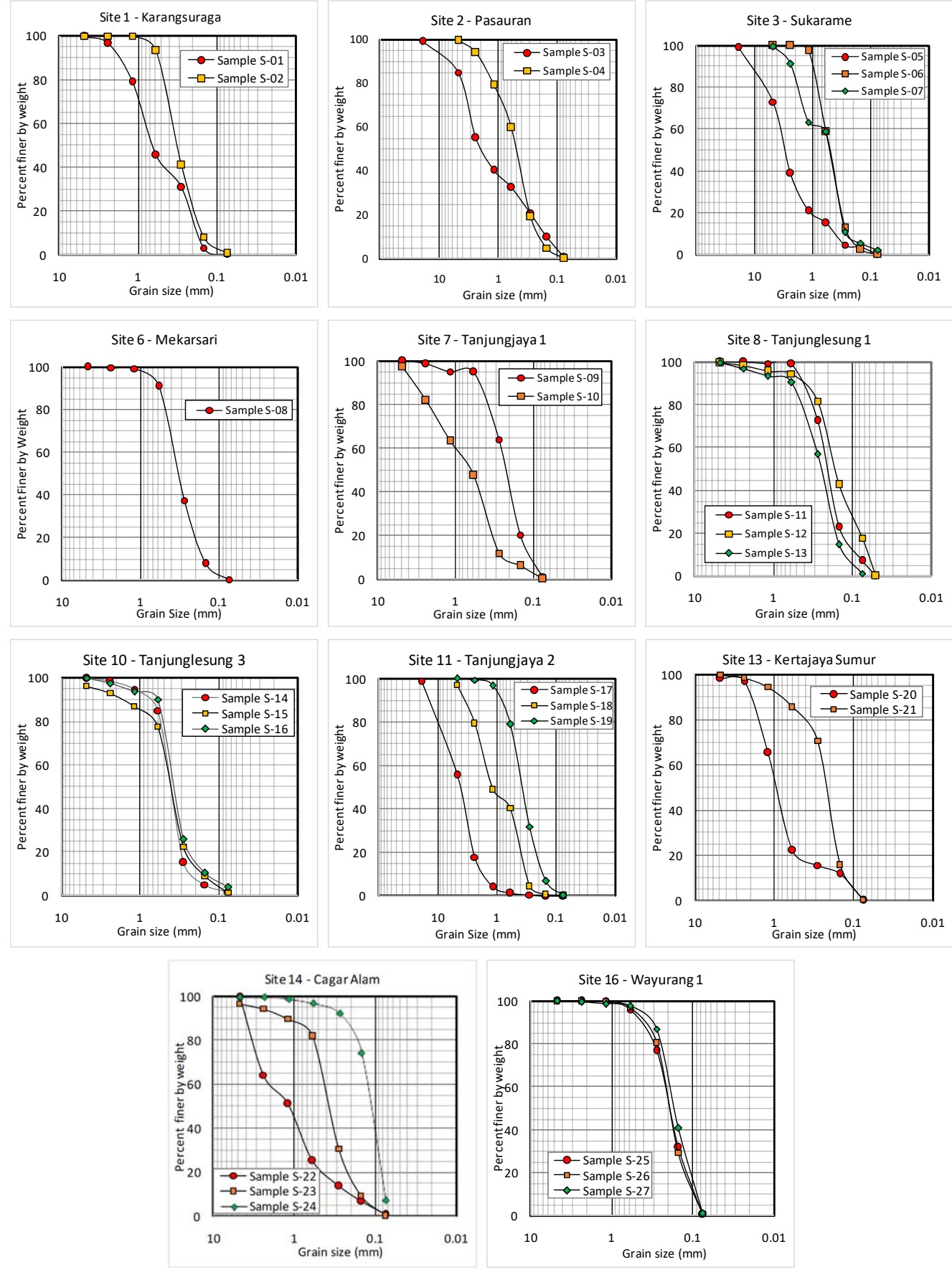

**Figure 8.** Sediment grain size results from sieve analysis for various sites.