# Peer review of "Runup, Inundation, and Sediment Characteristics of December 22 2018 Indonesia Sunda Strait Tsunami"

_Natural Hazards and Earth System Sciences, 2019_

## Referee Comment (RC1) · Anonymous Referee #1 · 24 Dec 2019

NHESS-2019-325: Runup, Inundation, and Sediment Characteristics of 22 December 2018 Indonesia Sunda Strait Tsunami

Authors: Wahyu Widiyanto, Wei-Cheng Lian, Shih-Chun Hsiao, Purwanto B. Santoso, Rudy T. Imananta

Overview:

Widiyanto et al. conducted field surveys of the 22 Dec 2018 Anak Krakatau volcano tsunami along the coastlines of Sunda Strait and reported wave runup distribution. They also collected sediment samples and performed tsunami deposit analysis.

[Figure]

I believe that this is an important study and the results are very useful. The manuscript reads well; its figures have good qualities and the structure of the manuscript is appropriate. However, I found some unclear points in the manuscript that needs to be corrected before publication. The details of runup survey are unclear and I made comments to help authors to correct it. Also the manuscript needs to compare its results with published papers on the Anak Krakatau tsunami and explain how this work connects with existing literature.

My recommendation is "Moderate Revision" with following comments. I encourage the authors to do the revisions quickly and resubmit soon in order to publish the paper earlier.

Comments:

Page 2, Line 13: please show two locations "Merak" and "Bakahueni" in Figure 1.

P2, L1-9: in this part of introduction, I think it would be very useful if you report the two recently published papers on the same event. They are:

Muhari, A., Heidarzadeh, M., Susmoro, H., Nugroho, H.D., Kriswati, E., Supartoyo, Wijanarto, A.B., Imamura, F., Arikawa, T. (2019). The December 2018 Anak Krakatau volcano tsunami as inferred from post-tsunami field surveys and spectral analysis. Pure and Applied Geophysics, https://doi.org/10.1007/s00024-019-02358-2.

Heidarzadeh, M., Ishibe, T., Sandanbata, O., Muhari, A., Wijanarto, A.B. (2020). Numerical modeling of the subaerial landslide source of the 22 December 2018 Anak Krakatoa volcanic tsunami, Indonesia. Ocean Engineering, 195, https://doi.org/10.1016/j.oceaneng.2019.106733.

You could say like this: "The numerical modelling of the Dec 2018 Anak Krakatau tsunami was performed by Heidarzadeh et al. (2020) while Muhari et al. (2019) conducted field surveys of this event to record tsunami runup along the coasts of Sunda Strait".

P3, L36: here please clarify which coastline? We have two coastlines which are High Tide Coastline (HTC) and Low Tide Coastline (LTC). You measured runup based on HTC or LTC? This is very important to clarify.

P3, L13-16: here you talk about runup measurements; but you do not explain about tidal level corrections. The tide level at the time of actual tsunami was different from tidal level at the time of surveys. Please explain about this and the corrections that you made.

P3, L34-40: please compare your runup heights with those of Muhari et al. (2019) [Pure and Applied Geophysics, https://doi.org/10.1007/s00024-019-02358-2] and explain why Muhari et al. reported maximum runup height of 13 m but you report max runup of 8? Is that because you did not survey same points? Please clarify.

P3, L34-40: Here also please compare your surveyed runup heights with published tide gauge records of Heidarzadeh et al. (2020) [Ocean Engineering, 195, https://doi.org/10.1016/j.oceaneng.2019.106733]. For example, your runup heights how many times are larger than tide gauge heights reported by Heidarzadeh et al.? this information can be very useful.

P4, L1: please show location "Sumur" in Figure 1.

P4, L3; 250 m. Add "m".

P4, L6: same comment as before for coastline; HTC or LTC?

P5, L27; how much is the value of gamma?

Figure 1: Please make the distance scale more clear and visible.

Figure 2: please increase fontsize. Most texts cannot be read.

Figure 3: please add name of each location after the letters "a", "b",....in each panel.

Figure 4; please add some location names in this figure'; for example location names

of 2, 4, 10 and 14.

Figure 5: Please add location name in each panel.

Figures 6 and 7: please combine these two figures to only one figure with two panels.

Figure 8: Please add location name in each panel.

Table 1: in column 3, please add time as well. You have only date now. What time of the day? This is very important because we can see how tidal status was at the time of your survey.

- Regards.

---

## Referee Comment (RC2) · Ahmet Cevdet Yalciner (Referee) · 8 Jan 2020

*Manuscript Number:* NHESS-2019-325

*Full Title:* Runup, Inundation, and Sediment Characteristics of December 22 2018 Indonesia Sunda Strait Tsunami

*Authors:* Wahyu Widiyanto, Wei-Cheng Lian, Shih-Chun Hsiao, Purwanto B. Santoso, Rudy T. Imananta

The authors conducted a field survey a month after the 22 December 2018 Anak Krakatau tsunami event. The paper presented and discussed the measurements of runup height, inundation distance, tsunami direction, and sediment characteristics at selected sites. The followings are my comments on the manuscript.

*Major Comments and Recommendations:*

*Page 1, Lines 8-9:* You had better rewrite the sentence "The affected area of the tsunami included a coastal area located at the edge of Sunda Strait, Indonesia." in such a way "The tsunami affected the coastal areas located at the edge of Sunda Strait, Indonesia."

*Page 1, Lines 13-14:* The sentence is grammatically incorrect. "Tsunami propagated radially from its source and arrived in coastal zone with direction was between 25° and 350° from North.". Please rewrite.

*Page 1, Lines 26-27:* There is an incorrect statement in the sentence "The southwestern slope of the mountain experienced a landslide below the sea surface that resulted in.." because the landslide not only occurred below the sea surface but also there is a subaerial part of the landslide.

*Page 2, Lines 12-15:* Any reference for such kind of information "It connects the two main islands of Java and Sumatra, whose population accounts for 79% of Indonesia's population. About 6.9 million people live in the coastal area of the strait in Banten Province and Lampung Province." OR "The strait, between Merak and Bakauheni, is the busiest inter-island crossing in Indonesia, with more than 50,000 passengers/day and more than 20,000 vehicles/day."

*Page 4, Lines 14-15:* "A relatively long inundation (284.2 m) was also found at Tanjungjaya 2, a site 15 with a relatively high runup." Any information on the steepness of the slope which can justify the situation given in this information?

*Page 6, Lines 12-13:* "We identified boulders moved by a tsunami wave and runup at three survey sites based on information from eyewitnesses and **their physical state."** The phrase in bold is redundant.

*Page 6, Lines 12-13:* "In addition, from the physical criteria given by Morton et al. (2007) and Paris et al. (2010), it was most likely that the boulders were moved by the tsunami." It is needed to mention a little bit about the "physical criteria" mentioned in this sentence and how you related it to your case.

*Page 6, Lines 33-34:* "and $\rho w$ is the density of sea water." Unit is missing. "The velocities were calculated **from Equation 3?** to be $u \geq 4.5$ m/s and $u \geq 4.0$ m/s for the 10.4-ton (Fig. 8a) and 9.4-ton (Fig. 8b) boulders, respectively." If so, please add the highlighted words.

*Page 7, Line 28:* "…and the direction of the tsunami was between 25° and 350° **from North.**" Better to add the highlighted words.

It is better to explain the reasons (local morphological conditions, ground material, ground slope etc.) of the discrepancies between theoretical deposit limit and the measured deposit limit at the locations where they do not fit well such as Sukarame, Tanjungjaya 1 and Cagar Alam.

Any information on the tidal situation of the area? Is there any detiding process performed on the measured values?

In conclusion part especially, why needed to use past tense for some findings? They are still valid. For example, "The largest boulder **had (has)** a diameter of 2.7 m and a weight of 10.4 tons. From the boulder movement, the tsunami velocity at the ground surface **was (is)** estimated to be more than 4.5 m/s. Sand size statistics **were (are)** also given in this report. The sediment grain size ranged from very fine sand to boulders, with medium sand (diameter: 0.25-0.5 mm) and coarse sand (diameter: 0.5 -1.0 mm) being dominant. All sediment samples tested in the laboratory **had (has)** a well sorted distribution, indicating that the grain sizes were relatively uniform.

**Figures and Typos:**

*Page 3, Line 13:* "terestrial" → "terrestrial"

*Figure 3:* Only places and arrows are shown in the pictures of Figure 3 which are not satisfactory for inferring the wave direction at these locations. Indication of the locations where each picture belongs to is necessary. Writing also the coordinates may be a good idea.

*Figure 4 and Page-4, Lines 27-35:* Can you please indicate the survey point IDs of the arrows shown in Figure 4 as well as the ones stated in these lines such as "Tanjung Lesung (sites 7-13)" or, for example, where is this Tanggamus area? Then, the statements in these lines on Page 4 will make sense while reading and looking at the figure.

*Page 4, Line 34:* "Table 1 contains the quantity of tsunami wave direction arrived in coastal area." Better rewrite this sentence in such a way "Tsunami wave direction from North arrived in coastal area is given/presented in Table 1 for the field survey sites."

*Page 4, Line 35:* "" → "North", please correct this type of typos throughout the manuscript.

*Page 5, Lines 3-4:* "Prehistoric (paleo-) tsunamis have been identified from sediment deposits (Atwater 1992; Dawson and Shi 2000; Peters, Jaffe and Gelfenbaum 2007)." Is this sentence a general statement since it is not clear if it is a general statement or mentioning about a specific study for a region for example? Better rewrite the sentence as "Prehistoric (paleo-) tsunamis have been identified from sediment deposits in several/many studies/publications (Atwater 1992; Dawson and Shi 2000; Peters, Jaffe and Gelfenbaum 2007).

*Page 5, Line 16:* "Four deposit pits were less than 50 m from the shoreline (11)." What is this 11 here?

*Page 5, Line 18:* "…and created a deposit a short distance from the…" → "…and created a deposit **at** a short distance from the…"

*Page 5, Line 23:* "reconstructing tsunami runup from sedimentary characteristics."

*Page 6, Line 24:* "Other smaller chunks also moved." → "Other smaller chunks **were** also moved."

---

## Author Comment (AC1) · 13 Jan 2020

Authors Response

Referee 1 – Anonymous

Overview: Widiyanto et al. conducted field surveys of the 22 Dec 2018 Anak Krakatau volcano tsunami along the coastlines of Sunda Strait and reported wave runup distribution. They also collected sediment samples and performed tsunami deposit analysis. I believe that this is an important study and the results are very useful. The manuscript reads well; its figures have good qualities and the structure of the manuscript is ap-

propriate. However, I found some unclear points in the manuscript that needs to be corrected before publication. The details of runup survey are unclear and I made comments to help authors to correct it. Also the manuscript needs to compare its results with published papers on the Anak Krakatau tsunami and explain how this work connects with existing literature. My recommendation is "Moderate Revision" with following comments. I encourage the authors to do the revisions quickly and resubmit soon in order to publish the paper earlier.

Response to overview: We would like to thank Referee 1 for encouraging comments and constructive suggestions towards improving our manuscript. We summarize comments from Referee 1, author's response, and author's changes in manuscript as follows. Changes in manuscript will be available in marked-up/revised manuscript if we have a chance to revise the manuscript.

Comment 1: Page 2, Line 13: please show two locations "Merak" and "Bakahueni" in Figure 1.

Response 1: Thanks for suggestion. Merak and Bakauheni are ferry ports with crowded traffic. They are important place to show. We add legends in Figure 1 in mark-up manuscript to show the two locations.

Comment 2: P2, L1-9: in this part of introduction, I think it would be very useful if you report the two recently published papers on the same event. They are: Muhari, A., Heidarzadeh, M., Susmoro, H., Nugroho, H.D., Kriswati, E., Supartoyo, Wijanarto, A.B., Imamura, F., Arikawa, T. (2019). The December 2018 Anak Krakatau volcano tsunami as inferred from post-tsunami field surveys and spectral analysis. Pure and Applied Geophysics, https://doi.org/10.1007/s00024-019-02358-2. Heidarzadeh, M., Ishibe, T., Sandanbata, O., Muhari, A., Wijanarto, A.B. (2020). Numerical modeling of the subaerial landslide source of the 22 December 2018 Anak Krakatoa volcanic tsunami, Indonesia. Ocean Engineering, 195, https://doi.org/10.1016/j.oceaneng.2019.106733. You could say like this: "The numerical modelling of the Dec 2018 Anak Krakatau

tsunami was performed by Heidarzadeh et al. (2020) while Muhari et al. (2019) conducted field surveys of this event to record tsunami runup along the coasts of Sunda Strait".

Response 2: Thanks for suggestion. We report the two papers in part of introduction, and add them in part of reference belongs to marked-up manuscript.

Change in manuscript: The numerical modelling of the December 2018 Anak Krakatau tsunami was performed by Heidarzadeh et al. (2020) while Muhari et al. (2019) conducted field surveys of this event to record tsunami runup along the coasts of Sunda Strait.

Comment 3: P3, L36: here please clarify which coastline? We have two coastlines which are High Tide Coastline (HTC) and Low Tide Coastline (LTC). You measured runup based on HTC or LTC? This is very important to clarify.

Response 3: We measure based on the coastline of measurement time. The numbers shown in the manuscript version 1 were original measurement values. Now, we correct them for tide using WXTide version 47 software. Tsunami arrival times are determined based on tidal record that show tsunami waveform. Four tidal gauge record were obtained from Geospatial Information Agency, Indonesia. They are Marina Jambu, Ciwandan, Panjang and Kota Agung. Or we can use the tide gauge data in article by Heidarzadeh (2020) which is published officially.

Change in manuscript: The runup was measured by determining the height difference between the highest point of sea water rise onto land and the coastline. Runup is influenced by the characteristics of the ground surface and slope. The measurement results from our field surveys show that runup ranged from 1 to 9 m (Table 1 and Fig. 1). The values in the table and figure has been corrected for tide to obtain elevation from sea level at time of tsunami.

Comment 4: P3, L13-16: here you talk about runup measurements; but you do not

explain about tidal level corrections. The tide level at the time of actual tsunami was different from tidal level at the time of surveys. Please explain about this and the corrections that you made.

Response 4: The numbers appear in the manuscript version 1 especially in Table 1 and Figure 1 were original numbers come from measurement. We have not corrected them for tide therefore we need to correct them using tidal levels. We use WXTide version 47 software to correct it. The station we use is Ciwandan, Serang, and Teluk betung tidal gauge station. The corrected values will be shown in mark-up manuscript if this process continue to next stage.

Change in manuscript: Measurements of runup and inundation were conducted using conservative terestrial surveying methods with optical and laser devices (e.g., total stations, handheld GPS devices, and laser distance meters). We measured run-up and inundation based on coastline at the time of survey. Run-up were corrected to calculate heights above sea level because the tide level at the time of actual tsunami was different from tidal level at the time of surveys. We use WXTide software version 4.7 for correcting elevation. Elevation values of each survey site were corrected with the nearest tidal gauge available. We used 3 tide station in Ciwandan, Labuhan and Teluk Betung.

Comment 5: P3, L34-40: please compare your runup heights with those of Muhari et al. (2019) [Pure and Applied Geophysics, https://doi.org/10.1007/s00024-019-02358-2] and explain why Muhari et al. reported maximum runup height of 13 m but you report max runup of 8? Is that because you did not survey same points? Please clarify.

Response 5: Thanks for recommendation. Yes right. The difference is because we measured in different points. Our maximum run-up point (Cagar Alam) is located very far from maximum run-up point (Tanjungjaya) belongs to Muhari et al. Actually, we also have a measurement point near Muhari et al. measured. It is site Tanjungjaya-2 or local people call it Cipenyu Beach. The height of run-up is 9 m, we add it in markedup manuscript and become the highest run-up in our survey. Nevertheless, this value is still significant different with Muhari et al. since we measure in flat valley part of Cipenyu Beach while Muhari et al. measured in hilly coast of Cipenyu Beach.

Change in manuscript: The measurement results from our field surveys show that run-up ranged from 1 to 9 m (Table 1 and Fig. 1). A runup height of about 1 m was found in many locations, at which no damage was found. The highest runup was found at the Tanjung Jaya 2, Cagar Alam, and Kunjir sites, with heights of 9.0, 7.8, and 7.7 m respectively. Site Tanjungjaya 2 is located in Cipenyu Beach. Muhari et al. (2019) reported maximum runup height of 13 m in area around Tanjungjaya/Cipenyu Beach as well. This value is significant different with our maximum run-up since we measure in flat valley part of Cipenyu Beach while Muhari et al. measured in hilly coast of Cipenyu Beach.

Comment 6: P3, L34-40: Here also please compare your surveyed runup heights with published tide gauge records of Heidarzadeh et al. (2020) [Ocean Engineering, 195, https://doi.org/10.1016/j.oceaneng.2019.106733]. For example, your runup heights how many times are larger than tide gauge heights reported by Heidarzadeh et al.? this information can be very useful.

Response 6: Thanks for the interesting recommendation. We compare our surveyed runup heights with published tide gauge records of Heidarzadeh et al. (2020) [Ocean Engineering, 195, https://doi.org/10.1016/j.oceaneng.2019.106733]. We use 3 tide gauges from the paper: Ciwandan, Marina Jambu and Panjang. Others (Kota Agung, Bengkurat, Binuangeun) are too far from our survey site. We read that maximum amplitudes at Ciwandan, Marina Jambu, and Panjang are 1.15 m, 2.8 m, and 1.25 m respectively. Ciwandan tide gauge is used to evaluate runup heights at site Karangsuraga, Pasauran, Sukarame and Pejamben. It results in average runup heights 4 times larger than amplitude at tide gauge heights. Note that the sites are relatively far from the tide gauge. Marina Jambu is used to evaluate runup heights at sites Sukamaju, Karangsari, Tanjungjaya 1, Tanjunglesung (1,2,3), Tanjungjaya 2, Banyuasih,

Kertajaya Sumur, Cagar Alam. It results in average runup heights 1.15 times larger than amplitude at tide gauge. Panjang tide gauge is used to evaluate sites of Bumiwaras, Wayurang 1, Wayurang 2, Kotaguring, Sukaraja, and Kunjir. It results in average runup heights of 3.1 times larger than amplitude at tide gauge.

Change in manuscript: Our surveyed runup heights are compared with published tide gauge records of Heidarzadeh et al. (2020). Three tide gauges from the article (Ciwandan, Marina Jambu and Panjang) are used. Maximum amplitudes at Ciwandan, Marina Jambu, and Panjang are 1.15 m, 2.8 m, and 1.25 m respectively. Ciwandan tide gauge is used to evaluate runup heights at site Karangsuraga, Pasauran, Sukarame and Pejamben. Marina Jambu tide gauge is used to evaluate runup heights at sites Sukamaju, Karangsari, Tanjungjaya 1, Tanjunglesung (1,2,3), Tanjungjaya 2, Banyuasih, Kertajaya Sumur, Cagar Alam. Besides, Panjang tide gauge is used to evaluate sites of Bumiwaras, Wayurang 1, Wayurang 2, Kotaguring, Sukaraja, and Kunjir. It is indicated that averaged runup heights of each site associated with the tide gauge are 4 times, 1.15 times, and 3.1 times larger than maximum amplitude at the Ciwandan, Marina Jambu, and Panjang respectively. The sites are relatively far from the tide gauge.

Comment 7: P4, L1: please show location "Sumur" in Figure 1.

Response 7: Alright, we show location Sumur in Figure 1 and will appear in marked-up manuscript.

Comment 8: P4, L6: same comment as before for coastline; HTC or LTC?

Response 8: We measured inundation distance based on the coastline of surveys time. The numbers shown in the manuscript version 1 were original measurement values. Correct values with tidal data will be shown in marked-up manuscript.

Change in manuscript: The distance from the runup point to the coastline is defined as the inundation distance (IOC Manuals and Guides No. 37, 2014). This distance can be easily obtained using a distance measurement instrument or GPS. We used total

station for this purpose. The coastlines elevation in our survey were corrected with tide elevation of several tide gauge in Sunda Strait. The results of our field measurements show that the inundation distance ranged from 10 to 290 m (Table 1 and Fig. 1). We

Comment 9: P4, L3; 250 m. Add "m".

Response 9: Thanks for thorough review, we add "m" in the value.

Change in manuscript: The topography is relatively flat but suddenly rises at a distance of about 250 m from the coastline due to a long hill.

Comment 10: how much is the value of gamma?

Response 10: Gamma is a comparison factor between uprush time and total uprush plus backwash time. In our paper, gamma varies from 0.03365 to 0.889192. We use some asumption, e.g. velocity of tsunami flow 5-6 m/s and period from the time of first wetting to final drying of inundated ground 2-5 hours. They depend on length of inundation and morphology.

Comment 11: Figure 1: Please make the distance scale more clear and visible.

Response 11: OK, thanks. We modify it in order to be visible and clearer. It will be ready in marked-up manuscript.

Comment 12: Figure 2: please increase fontsize. Most texts cannot be read.

Response 12: Alright, we increase the font size in order to be readable. It will ready in marked-up manuscript.

Comment 13: Figure 3: please add name of each location after the letters "a", "b",: : :.in each panel.

Response 13: Thanks for your suggestion to make the figure clearer. Location name for a is Carita Beach ; b = Tanjung Lesung; c = Cagar Alam; d = Cagar Alam; e = Tanjung Lesung; f = Tanjung Lesung; g = Cagar Alam.

[Figure]

Comment 14: Figure 4; please add some location names in this figure'; for example location names of 2, 4, 10 and 14.

Response 14: Thanks. Actually, there are location names in Figure 4, but they are not readable. We make them more visible and we add other location names and site number to Figure 4. Location name of 2 is Pasauran; 4 = Pejamben; 10 = Tanjung Lesung; 14 = Cagar Alam.

Comment 15: Figure 5: Please add location name in each panel.

Response 15: Thanks for your suggestion. We add the location name to Figure 5 while the coordinates of the test pits can be seen in Table 2. The location name of upper left panel is Cagar Alam, upper right is Sukarame, lower left is Karangsuraga, and lower right is Cagar alam. These name will appear in marked-up manuscript.

Comment 16:: Figures 6 and 7: please combine these two figures to only one figure with two panels.

Response 16: Thanks for your suggestion. One figure with two panels will make the manuscript more effective and efficient. We combine Figure 6 and 7 and will be ready in marked-up manuscript.

Comment 17: Table 1: in column 3, please add time as well. You have only date now. What time of the day? This is very important because we can see how tidal status was at the time of your survey.

Response 17: We recorded the times of survey for each site. We add them in column 3 of Table 1 in marked-up manuscript.

---

## Author Comment (AC2) · 13 Jan 2020

Authors Response

Referee 2 – Prof. Ahmed Cevdet Yalciner The authors conducted a field survey a month after the 22 December 2018 Anak Krakatau tsunami event. The paper presented and discussed the measurements of runup height, inundation distance, tsunami direction, and sediment characteristics at selected sites. The followings are my comments on the manuscript.

Major comment and recommendation: Page 1, Lines 8-9: You had better rewrite the
sentence "The affected area of the tsunami included a coastal area located at the edge of Sunda Strait, Indonesia." in such a way "The tsunami affected the coastal areas located at the edge of Sunda Strait, Indonesia."

Response 1: Thanks for correction. We change the sentence according to your recommendation.

Change in manuscript: A tsunami caused by a flank collapse of the southwest part of the Anak Krakatau volcano occurred on December 22, 2018. The tsunami affected the coastal areas located at the edge of Sunda Strait, Indonesia. To gain an understanding of the tsunami event, field surveys were conducted a month after the incident.

Page 1, Lines 13-14: The sentence is grammatically incorrect. "Tsunami propagated radially from its source and arrived in coastal zone with direction was between 25° and 350° from North.". Please rewrite.

Response 2: Thanks for correction.

Change in manuscript: The tsunamis propagated radially from Anak Krakatau and reached the coastal zone with direction between 25° and 350° from North.

Page 1, Lines 26-27: There is an incorrect statement in the sentence "The southwestern slope of the mountain experienced a landslide below the sea surface that resulted in.." because the landslide not only occurred below the sea surface but also there is a subaerial part of the landslide.

Response 3: Thanks for the correction. We remove "below the sea surface".

Change in manuscript: The southwestern slope of the mountain experienced a landslide that resulted in the movement of sea water, which propagated to land in the form of a tsunami wave.

Page 2, Lines 12-15: Any reference for such kind of information "It connects the two main islands of Java and Sumatra, whose population accounts for 79% of Indonesia's

population. About 6.9 million people live in the coastal area of the strait in Banten Province and Lampung Province." OR "The strait, between Merak and Bakauheni, is the busiest inter-island crossing in Indonesia, with more than 50,000 passengers/day and more than 20,000 vehicles/day."

Response 4: We add the references for this part: BPS-Statistics Indonesia: Statistical Yearbook of Indonesia 2019, Jakarta, 2019. BPS-Statistics of Banten Province: Banten Province in Figures 2019, Serang, 2019. BPS-Statistics of Lampung Province: Lampung Province in Figures 2019, Bandar Lampung, 2019. Dirjen Perhubungan Darat: Perhubungan Darat dalam Angka 2018, Jakarta, 2019 Soeriaatmadja, W.: Indonesia plans traffic system for busy Sunda Strait, available at https://www.straitstimes.com/asia/se-asia/indonesia-plans-traffic-system-for-busy-sunda-strait (last acces, 10 January 2020), 2020.

Change in manuscript: It connects the two main islands of Java and Sumatra, whose population accounts for 79% of Indonesia's population (BPS-Statistics Indonesia, 2019). About 6.9 million people live in the coastal area of the strait in Banten Province and Lampung Province (BPS-Statistics of Banten Province, 2019; BPS-Statistics of Lampung Province, 2019). The strait, between Merak and Bakauheni, is the busiest inter-island crossing in Indonesia, with 17.824.392 passengers and 4.218.548 vehicles in 2018 (Dirjen Perhubungan Darat, 2019). The strait is also an international route for large ships. It is the second-most crowded waterway after Malacca Strait, with 70,000 vessels a year passing it (Soeriaatmadja, 2016).

Page 4, Lines 14-15: "A relatively long inundation (284.2 m) was also found at Tanjungjaya 2, a site 15 with a relatively high runup." Any information on the steepness of the slope which can justify the situation given in this information?

Response 5: Site Tanjungjaya 2 has different characteristics from other sites. This site is in the form of a valley plain with a small stream. Slope in the valley is relatively flat and suddenly changes sharply in hilly areas within 250-300 m from coastline. The runup point that we recorded is located on the slope change from mild to steep. These flat and sharp areas have slopes of approximately 0.025 and 0.06, respectively. Local people call this area as Cipenyu Beach. This is a sandy beach flanked by cliffs or hilly beaches.

Change in manuscript: A relatively long inundation (284.2 m) was also found at Tanjungjaya 2, a site 15 with a relatively high runup. This site is in the form of a valley plain with a small stream. Slope in the valley is relatively flat and suddenly changing steeply in hilly areas within 250-300 m from coastline. The run-up point we recorded is located on the slope change from mild to steep. These mild and steep areas have slopes of approximately 0.025 and 0.06, respectively. Local people call this area as Cipenyu Beach. This is a sandy beach flanked by cliffs or hilly beaches. Fortunately, not many people live around this site other than at a resort complex, which suffered severe damage.

Page 6, Lines 12-13: "We identified boulders moved by a tsunami wave and runup at three survey sites based on information from eyewitnesses and their physical state." The phrase in bold is redundant.

Response 6: Thanks for the correction. We delete the phrase in bold.

Change in manuscript: We identified boulders moved by a tsunami wave and runup at three survey sites based on information from eyewitnesses.

Page 6, Lines 12-13: "In addition, from the physical criteria given by Morton et al. (2007) and Paris et al. (2010), it was most likely that the boulders were moved by the tsunami." It is needed to mention a little bit about the "physical criteria" mentioned in this sentence and how you related it to your case.

Response 7: Thanks for recommendation. We conclude that the boulder was transported by tsunami mainly based on eyewitness information. We add a little bit about the "physical criteria" by Morton et al. (2007) and Paris et l. (2010) as we write in

change manuscript here.

Change in manuscript: Eyewitnesses said that these boulders were in new positions after the tsunami. In addition, from the physical criteria given by Morton et al. (2007) and Paris et al. (2010), it was most likely that the boulders were moved by the tsunami. One of criteria by Morton et al. (2007) we found in this site is a relatively thin (average < 25 cm) bed composed of normally graded sand consisting of a single structureless bed or a bed with only a few thin layers. Sediment thickness around the boulder is very thin. Paris et al. (2010) reported regarding boulder and fine sediment transport and deposition by the 2004 tsunami that most of the sediments deposited on land came from offshore, from fine sands to coral boulders, and with very high values of shear velocity (>30 cm/s). The boulder we found came from nearshore and a part of the boulder was submerged. We estimate that high shear velocity should occure to transport it. It was most possible by 22 December 2018 tsunami.

Page 6, Lines 33-34: "and w is the density of sea water." Unit is missing. "The velocities were calculated from Equation 3? to be $u \geq 4.5$ m/s and $u \geq 4.0$ m/s for the 10.4-ton (Fig. 8a) and 9.4-ton (Fig. 8b) boulders, respectively." If so, please add the highlighted words.

Response 8: Thanks for correction and suggestion. Unit for density of sea water is kg/m3. Right, the velocities were calculated from Equation 3. We add the highlighted words.

Change in manuscript: where $\mu$ is the friction coefficient, m is the boulder mass (kg), g is the gravitational acceleration, Cd is the drag coefficient, An is the area of the boulder projected normal to the flow (m2), and w is the density of sea water (kg/m3). The velocities were calculated from Equation 3 to be $u \geq 4.5$ m/s and $u \geq 4.0$ m/s for the 10.4-ton (Fig. 8a) and 9.4-ton (Fig. 8b) boulders, respectively.

Page 7, Line 28: "...and the direction of the tsunami was between 25° and 350° from North." Better to add the highlighted words.

Response 9: Thanks for the correction.

Change in manuscript: The survey results revealed that the runup height ranged from 1 to 8 m, the inundation distance was 10 to 300 m, and the direction of the tsunami was between 25° and 350° from North.

It is better to explain the reasons (local morphological conditions, ground material, ground slope etc.) of the discrepancies between theoretical deposit limit and the measured deposit limit at the locations where they do not fit well such as Sukarame, Tanjungjaya 1 and Cagar Alam.

Response 10: Thanks for your suggestion. The three location have morphological conditions may not ideal for applying the theoritical approach. Sukarame has beach scarp and tsunami flows across a stream around 90 m from coastline. Tanjungjaya 1 has also beach scarp and there is a sea wall (although not so high) that may block the sediment movement . Eventhough Tanjungjaya 1 has abundant material, low amplitude tsunami caused a few sand transport. Cagar alam has a relative bigger stream than Sukarame. In addition, Cagar Alam has dense vegetation since it is a national park.

Change in manuscript: Fig. 7 shows the distance of measured sediment deposition and water runup compared to the distance of theoretical sediment deposition calculated using Eq. 1 and Eq. 2, the results are in good agreement. However, Sukarame, Tanjungjaya 1 and Cagar Alam do not fit well. The three location have morphological conditions may not ideal for applying the theoritical approach. Sukarame has beach scarp and tsunami flows across a stream around 90 m from coastline. Tanjungjaya 1 has also beach scarp and there is a sea wall, although not so high, that may block the sediment movement. Eventhough Tanjungjaya 1 has abundant material, low amplitude tsunami caused a few sand transport. Cagar Alam has a relative bigger stream than Sukarame. In addition, Cagar Alam has dense vegetation since it is a national park. The distance of area with significant sediment deposits caused by the tsunami from the coast varied in the range of 15-200 m (average: 93 m) from the shoreline or 40%-90%

(average: 67%) of the inundation distance.

Any information on the tidal situation of the area? Is there any detiding process performed on the measured values?

Response 11: Yes, we have 4 tidal gauge data giving information on the tidal situation from the area. Our measured values shown in the manuscript version 1 are original values. We have not corrected the measured values in the manuscript version 1. We will show corrected values in mark-up manuscript if it continues to next stage. We use WXTide 47 software to correct the measured values. Tsunami arrival times are determined based on tidal record that show tsunami waveform. Four tidal gauge record were obtained from Geospatial Information Agency, Indonesia. They are Marina Jambu, Ciwandan, Panjang and Kota Agung.

Change in manuscript: Measurements of runup and inundation were conducted using conservative terestrial surveying methods with optical and laser devices (e.g., total stations, handheld GPS devices, and laser distance meters). We measured run-up and inundation based on coastline at the time of survey. Run-up were corrected to calculate heights above sea level at the time of the survey using WXTide software version 4.7. Elevation values of each survey site were corrected with the nearest tidal gauge available. We used 3 station in Ciwandan, Labuhan and Teluk Betung, for corrections.

In conclusion part especially, why needed to use past tense for some findings? They are still valid. For example, "The largest boulder had (has) a diameter of 2.7 m and a weight of 10.4 tons. From the boulder movement, the tsunami velocity at the ground surface was (is) estimated to be more than 4.5 m/s. Sand size statistics were (are) also given in this report. The sediment grain size ranged from very fine sand to boulders, with medium sand (diameter: 0.25-0.5 mm) and coarse sand (diameter: 0.5 -1.0 mm) being dominant. All sediment samples tested in the laboratory had (has) a well sorted distribution, indicating that the grain sizes were relatively uniform.

Response 12: Thanks for correction regarding basic writing. We check throughout
manuscript about it and revise as you suggest.

Change in manuscript: The largest boulder has a diameter of 2.7 m and a weight of 10.4 tons. From the boulder movement, the tsunami velocity at the ground surface is estimated to be more than 4.5 m/s. Sand size statistics are also given in this report. The sediment grain size ranged from very fine sand to boulders, with medium sand (diameter: 0.25-0.5 mm) and coarse sand (diameter: 0.5 -1.0 mm) being dominant. All sediment samples tested in the laboratory has a well sorted distribution, indicating that the grain sizes are relatively uniform.

Figures and Typos:

Page 3, Line 13: "terestrial" → "terrestrial"

Response 13: Thanks.

Change in manuscript: Measurements of runup and inundation were conducted using conservative terrestrial surveying methods with optical and laser devices (e.g., total stations, handheld GPS devices, and laser distance meters).

Figure 3: Only places and arrows are shown in the pictures of Figure 3 which are not satisfactory for inferring the wave direction at these locations. Indication of the locations where each picture belongs to is necessary. Writing also the coordinates may be a good idea.

Response 14: Thanks for comment and recommendation. We add the site names and coordinates. Fortunately, we recorded coordinates for each locations, for instance shown by Fig. 3a, a man was recording a coordinates on a fallen tree. Fig 3a (105.829587° , -6.316732°) Pejamben Fig 3b (105.652357° , -6.481177°) Tanjunglesung Fig 3c 105.378817° , -6.674535° Cagar Alam Fig 3d 105.378692° , -6.676228° Cagar Alam Fig 3e 105.830286° , -6.316416° Pejamben Fig 3f 105.829155° , -6.317243° Pasauran Fig 3g 105.830011° , -6.316646° Pejamben Fig 3h 105.379027° , -6.675038° Cagar Alam We add them to mark-up manuscript

Figure 4 and Page-4, Lines 27-35: Can you please indicate the survey point IDs of the arrows shown in Figure 4 as well as the ones stated in these lines such as "Tanjung Lesung (sites 7-13)" or, for example, where is this Tanggamus area? Then, the statements in these lines on Page 4 will make sense while reading and looking at the figure.

Response 15: Yes, we can put the point IDs for each arrows in Fig 4. Also add the position of location mentioned in Page 4 but not shown in Fig 4 such as Tanggamus, Sertung island and Indian Ocean. Thanks for kind recommendation.

Change in manuscript: Figure 4 revised.

Page 4, Line 34: "Table 1 contains the quantity of tsunami wave direction arrived in coastal area." Better rewrite this sentence in such a way "Tsunami wave direction from North arrived in coastal area is given/presented in Table 1 for the field survey sites."

Response 16: Thanks for your kindly correction.

Change in manuscript: Tsunami wave direction from North arrived in coastal area is given in Table 1 for the field survey sites.

Page 4, Line 35: "north" → "North", please correct this type of typos throughout the manuscript.

Response 17: We correct them throughout the manuscript.

Change in manuscript: "north" → "North" Page 4 Line 30 need to revise Page 4 Line 35 need to revise Page 1 Line 14 already correct Table 1 already correct

Page 5, Lines 3-4: "Prehistoric (paleo-) tsunamis have been identified from sediment deposits (Atwater 1992; Dawson and Shi 2000; Peters, Jaffe and Gelfenbaum 2007)." Is this sentence a general statement since it is not clear if it is a general statement or mentioning about a specific study for a region for example? Better rewrite the sentence as "Prehistoric (paleo-) tsunamis have been identified from sediment deposits in several/many studies/publications (Atwater 1992; Dawson and Shi 2000; Peters, Jaffe and Gelfenbaum 2007).

Response 18: Thanks for the suggestion. This is a general statement regarding studies on identification of prehistoric (paleo-) tsunamis from sediment deposits.

Change in manuscript: Prehistoric (paleo-) tsunamis have been identified from sediment deposits in several studies (Atwater 1992; Dawson and Shi 2000; Peters, Jaffe and Gelfenbaum 2007).

Page 5, Line 16: "Four deposit pits were less than 50 m from the shoreline (11)." What is this 11 here?

Response 19: Thanks for very thorough review. (11) should be (Figure 6). We used "cross-reference" menu in MS Word but it cause problem in process of converse to PDF file. We repair it.

Change in manuscript: Four deposit pits were less than 50 m from the shoreline (Figure 6).

Page 5, Line 18: "...and created a deposit a short distance from the..." → "...and created a deposit at a short distance from the..."

Response 20: Thanks for very thorough review.

Change in manuscript: Another was at site 13 (Kertajaya Sumur), where high-density housing blocked the sediment transport and created a deposit at short distance from the shoreline.

Page 5, Line 23: "reconstructing tsunamis runup from sedimentary characteristics."

Response 21: Thanks for very thorough review.

Change in manuscript: Soulsby et al. (2007) proposed a mathematical model for reconstructing tsunami runup from sedimentary characteristics.

Page 6, Line 24: "Other smaller chunks also moved." → "Other smaller chunks were also moved."

Response 22: Thanks for correction.

Change in manuscript: Other smaller chunks were also moved.

We would like to thank Prof Ahmed Cevdet Yalciner (Referee 2) for the constructive comments, very detail corrections, and recommendations towards improving our manuscript. We are improving our writing quality based on your kind suggestion. These comments are all valuable and very helpful for improving our paper. We appreciate that we have a chance to revise the manuscript as you recommend and to resubmit our manuscript will meet your approval.